# Revisiting LLM Reasoning via Information Bottleneck

## Abstract

Large language models (LLMs) have recently demonstrated remarkable progress in reasoning capabilities through reinforcement learning with verifiable rewards (RLVR). By leveraging simple rule-based rewards, RL effectively incentivizes LLMs to produce extended chain-of-thought (CoT) reasoning trajectories, progressively guiding them toward correct answers. However, existing approaches remain largely heuristic and intuition-driven, limiting the development of principled methodologies. In this paper, we present a theoretical characterization of LLM reasoning grounded in information bottleneck (IB) principle, introducing *IB-aware reasoning optimization* (IBRO), a framework that encourages reasoning trajectories to be both *informative* about the final correct answer and *generalizable* across diverse prompts. We derive a practical token-level surrogate objective and propose an efficient approximation, resulting in the lightweight *IB regularization* method. This technique integrates seamlessly into existing RL-based post-training frameworks without additional computational overhead, requiring only a one-line code modification. Empirically, we validate IB regularization across multiple mathematical reasoning benchmarks and RL algorithms, demonstrating consistent improvements in LLM reasoning performance.

## 1 Introduction

Benefiting from the extensive knowledge embedded in pre-trained large language models (LLMs), post-training has increasingly focused on reinforcement learning (RL) as a means to further enhance human alignment (Ouyang et al., 2022) and reasoning capabilities (Jaech et al., 2024). Recent studies have demonstrated that reinforcement learning with verifiable rewards (RLVR) offers a scalable and effective approach for incentivizing reasoning in LLMs (Guo et al., 2025). Unlike approaches that rely on fine-grained process-level supervision or chain-of-thought (CoT) annotations, RLVR leverages easily verifiable, rule-based outcome rewards. This training paradigm encourages models to spontaneously generate coherent reasoning chains, leading to substantial improvements on challenging tasks in coding and mathematics.

Although recent empirical advances are promising, approaches for enhancing LLM reasoning remain largely heuristic and intuition-driven, hindering the development of principled methodologies. For instance, exploration, a core characteristic of RL, plays a critical role in discovering high-quality reasoning trajectories and is typically measured by generation uncertainty, *i.e.*, entropy. Since entropy collapse is frequently observed during LLM post-training (He et al., 2025; Liu et al., 2025b), prior works often advocate heuristically maintaining high generation entropy, *i.e.*, encouraging token-level uncertainty, during post-training (Cui et al., 2025; Cheng et al., 2025; Yao et al., 2025). In contrast, another line of research suggests that explicitly reducing entropy or uncertainty, even in the absence of reward signals, can lead to improved reasoning performance (Agarwal et al., 2025; Gao et al., 2025; Li et al., 2025). These conflicting findings underscore the need for a rigorous theoretical understanding of reasoning in LLMs, which remains elusive yet essential.

In this work, we address this theoretical gap by analyzing LLM reasoning from the perspective of information bottleneck (IB) principle (Tishby et al., 2000; Tishby & Zaslavsky, 2015), which emphasizes the importance of discarding irrelevant information while preserving task-relevant signals. We introduce *IB-aware reasoning optimization* (IBRO), an information-theoretic framework designed to optimize LLM reasoning capability (Definition 1). Specifically, IBRO encourages reasoning

processes to maximize informativeness with respect to (*w.r.t.*) correct answers while minimizing dependency on irrelevant, prompt-specific details. We then derive a token-level surrogate IBRO objective (Theorem 1). To facilitate practical implementation, we derive an efficient approximation of the IBRO objective, resulting in a novel *IB regularization* term. Concretely, IB regularization modulates the token-level entropy based on their corresponding advantages, incentivizing higher entropy for critical tokens and penalizing uninformative ones. Our IB regularization seamlessly integrates into existing RL-based post-training frameworks, introducing negligible computational overhead and requiring only a single line of code modification.

To comprehensively evaluate the effectiveness of our IB regularization, we conduct LLM post-training with two representative RL algorithms: PPO (Schulman et al., 2017) and DAPO (Yu et al., 2025), which represent the mainstream critic-based and critic-free paradigms, respectively (Shao et al., 2024). All experiments are conducted on Qwen2.5-7B (Yang et al., 2024), a widely adopted LLM base model that has not been specifically optimized for instruction following or reasoning. We assess performance on multiple mathematical reasoning benchmarks, including AMC23, AIME24, and AIME25 (Hendrycks et al., 2021). Across these tasks, IB regularization yields consistent and stable improvements, with an average gain of two points on both PPO and DAPO. We further provide fine-grained analysis *w.r.t.* entropy dynamics and response length, demonstrating the stability and compatibility of IB regularization.

In sum, our main contributions are as follows:

- We introduce IBRO, an information-theoretic framework grounded in the information bottleneck principle that characterizes LLM reasoning.
- We propose a computationally efficient IB regularization method based on token-level advantage, enabling seamless integration into existing RL-based training pipelines.
- We conduct extensive experiments on mathematical reasoning benchmarks with multiple RL algorithms, showing that IB regularization consistently improves performance while incurring minimal computational overhead.

## 2 RELATED WORKS

**LLM Reasoning** Pre-training endows LLMs with vast amounts of knowledge, and a key technique for further enhancing their reasoning capabilities is CoT prompting, which encourages step-by-step problem solving (Wei et al., 2022). Recent seminal works such as OpenAI-o1 (Jaech et al., 2024) and DeepSeek-R1 (Guo et al., 2025) adopt RL post-training to incentivize the emergence of CoT, enabling models to tackle complex reasoning tasks such as mathematics and code generation. Building on these efforts, a growing body of research has sought to understand the key factors in RL post-training that contribute to improved reasoning. One critical factor is generation entropy, which quantifies the uncertainty in the model's token-level output distribution. In practice, entropy often collapses rapidly toward zero during post-training, leading to overconfident predictions, reduced exploration, and ultimately, diminished gains in reasoning ability. To counter this, Yu et al. (2025) propose ClipHigher that relaxes clip constraints to allow more off-policy update for low-probability tokens. In parallel, explicit entropy regularization has gained attention as a direct intervention. However, its efficacy remains debated: on one hand, entropy minimization has been shown to promote reasoning without relying on explicit reward signals (Agarwal et al., 2025; Gao et al., 2025; Li et al., 2025); on the other hand, several works advocate for maintaining higher entropy to preserve exploration and thus foster reasoning (Cui et al., 2025; Cheng et al., 2025; Yao et al., 2025). In this paper, we revisit the challenge of LLM reasoning from an information-theoretic perspective, aiming to balance informativeness and generalization in reasoning process. Our analysis leads to a simple yet effective advantage-aware entropy regularization, which integrates seamlessly into existing RL post-training methods.

**Information-theoretical LLM** A growing body of work aims to understand and improve LLM training and inference through information-theoretic principles. Ton et al. (2025) identify critical CoT steps by estimating information gain via a supervised fine-tuning model, while Wang et al. (2025) similarly employ step-wise information gain as a dense reward for LLM fine-tuning. Chang (2024) further demonstrate that pairing a high-entropy exploratory model with a low-entropy stable model can yield more robust multi-LLM reasoning.

Within information theory, the Information Bottleneck (IB) principle posits that an effective latent representation should (1) discard irrelevant information from the input to promote generalization, achieved by minimizing the mutual information between the input and the latent features, (2) while retaining information that is predictive of the target, achieved by maximizing the mutual information between the latent code and the label (Tishby et al., 2000; Tishby & Zaslavsky, 2015). Although computing mutual information terms is generally intractable, Alemi et al. (2017) propose a variational lower bound that enables practical estimation. IB principle has received empirical support from Saxe et al. (2018) and theoretical justification from Kawaguchi et al. (2023). Yang et al. (2025b) apply IB to analyze how LLMs process task-relevant information by directly measuring hidden-state representations. In this work, we revisit LLM reasoning through the lens of the IB principle and derive a simple yet effective regularization term to enhance reasoning quality. While Yu (2025) also analyze LLMs from an IB perspective, their focus lies in the pre-training phase, where they minimize matrix-based entropy (Giraldo et al., 2014) for compression. By contrast, we target the post-training stage and introduce a novel IB-based formulation specifically designed to enhance LLM reasoning.

## 3 PRELIMINARIES

**RLVR** Given a question or prompt $q$, the LLM $\pi_\theta$, parameterized by $\theta$, generates a response as a sequence of tokens $(o_1, o_2, \ldots, o_T)$ in an autoregressive manner, where each token is sampled from $\pi_\theta(o_t \mid o_{<t}, q)$. In the RLVR setting, the training dataset is $\mathcal{S} = \{(q_i, a_i)\}_{i=1}^m$, where $a_i$ denotes the ground-truth answer to question $q_i$, without intermediate reasoning chains. When prompting the LLM with $q_i$, a binary reward $R \in \{0, 1\}$ can be directly assigned by matching the final answer in the generated response with $a_i$. For simplicity, we omit other rule-based rewards such as format compliance or length penalties. Several RL objectives have been developed based on PPO (Schulman et al., 2017). For completeness, we briefly recall the standard PPO objective:

$$\mathcal{J}_{\text{PPO}} = \mathbb{E}_{(q,a)\sim\mathcal{S},\, o_{\leq t}\sim\pi_{\theta_{\text{old}}}(\cdot|q)} \left[ \min\left(s_t A_t,\, \text{clip}\left(s_t, 1-\epsilon, 1+\epsilon\right) A_t\right)\right],$$

where $\pi_{\theta_{\text{old}}}$ is the sampling policy, $s_t = \frac{\pi_\theta(o_t|o_{<t}, q)}{\pi_{\theta_{\text{old}}}(o_t|o_{<t}, q)}$ is the importance sampling ratio, $A_t = A(o_t; o_{<t}, q)$ denotes the advantage of selecting token $o_t$, and the hyperparameter $\epsilon$ controls the clipping range. Intuitively, $A_t$ quantifies how much better token $o_t$ is compared to other possible tokens at position $t$.

While PPO typically requires training a separate critic model to estimate $A_t$ for each token, this becomes challenging in the LLM regime with large vocabularies. Recent methods such as GRPO (Shao et al., 2024) eliminate the need for critic learning by introducing a group-relative advantage strategy. Specifically, for each question $q$, $G$ rollouts are generated by $\pi_{\theta_{\text{old}}}$, and their corresponding rewards are denoted by $\{R_i\}_{i=1}^G$. These rewards are then normalized to compute token-level advantages for the $i$-th response:

$$A_{i,t} = \frac{R_i - \text{mean}\left(\{R_1, R_2, \cdots, R_G\}\right)}{\text{std}\left(\{R_1, R_2, \cdots, R_G\}\right)}.$$

This formulation implies that all tokens within the same response share the same advantage under the critic-free paradigm.

**Mutual Information (MI)** measures the shared information between two random variables:

$$I(X; Y) = H(X) - H(X \mid Y) = H(Y) - H(Y \mid X),$$

where the entropy $H(\cdot)$ is defined as $H(X) = \mathbb{E}_{X\sim p(X)}\left[-p(X)\log p(X)\right]$. For notational simplicity, we slightly abuse notation by letting $X$ and $Y$ denote both random variables and specific realizations. A larger $I(X; Y)$ indicates stronger statistical dependence between $X$ and $Y$; in other words, knowing one substantially reduces the uncertainty of the other.

**Information Bottleneck (IB)** offers a principled framework for balancing a model's *representation complexity* and its *predictive power*. Given a model $\mathcal{M}$ that encodes the input $X$ into a latent representation $Z$, which is subsequently used to predict the target $Y$, the IB principle seeks an optimal representation $Z$ by solving

$$\min_{Z\sim\mathcal{M}(Z|X)} I(X; Z) - \beta I(Z; Y),$$

where $I(X; Z)$ quantifies the information preserved about the input, reflecting the complexity of the representation, and $I(Z; Y)$ measures the predictive content of $Z$. The coefficient $\beta > 0$ balances compression against predictive accuracy. Minimizing $I(X; Z)$ discourages the retention of spurious details, thereby enhancing generalization, while maximizing $I(Z; Y)$ ensures that the representation remains informative for the prediction task. Together, the IB objective promotes representations that are both compact and task-relevant.

## 4 REASONING VIA INFORMATION BOTTLENECK

Given a powerful post-trained LLM $\pi$ and a prompt $\boldsymbol{q}$, the LLM engages in a reasoning process to generate a CoT $\boldsymbol{r}$, with the ultimate goal of producing the correct answer $\boldsymbol{a}$. Here, $\boldsymbol{a}$ denotes the ground truth rather than LLM predictions. This raises a fundamental question: *what characterizes a good reasoning process, or equivalently, a good CoT?* At a high level, a desirable CoT should be (1) *informative*, effectively guiding the model toward the correct answer; and (2) *generalizable* such that the reasoning process does not rely excessively on the specific prompt and thus transfer well to unseen questions. Building on the information bottleneck principle, we propose the following formulation to optimize LLM reasoning in an IB-aware manner.

**Definition 1** (IB-Aware Reasoning Optimization). *Given a base LLM $\pi$ and a dataset of prompt-answer pairs $(\boldsymbol{q}, \boldsymbol{a})$, we optimize the reasoning ability of $\pi$ by*

$$\min_{\pi(\boldsymbol{r}|\boldsymbol{q})} I(\boldsymbol{q}; \boldsymbol{r}) - \beta I(\boldsymbol{r}; \boldsymbol{a}),$$

*where $I(\boldsymbol{q}; \boldsymbol{r})$ quantifies the information retained from the prompt and $I(\boldsymbol{r}; \boldsymbol{a})$ measures the informativeness of the reasoning path toward the answer.*

IB-aware reasoning optimization (IBRO) seeks reasoning processes $\boldsymbol{r}$ that minimize dependence on unnecessary details in $\boldsymbol{q}$, while maximizing relevance to the target answer $\boldsymbol{a}$, and the hyperparameter $\beta > 0$ balances compression and predictiveness.

**Remark 1.** *Since overfitting is rarely a concern in the context of LLM RL post-training, response accuracy is typically prioritized over compression. As a result, a large value of $\beta > 1$ is preferred to bias the optimization toward maximizing accuracy rather than generalization.*

### 4.1 PRACTICAL OBJECTIVE

While IBRO offers a principled guideline for LLM reasoning optimization, the mutual information terms are intractable and do not naturally align with the token-level training objectives commonly used in LLM fine-tuning. To this end, we derive a more practical objective suitable for both understanding and implementation. First, the mutual information terms can be expressed by entropy as

$$I(\boldsymbol{q}; \boldsymbol{r}) = H(\boldsymbol{r}) - H(\boldsymbol{r} \mid \boldsymbol{q}), \quad I(\boldsymbol{r}; \boldsymbol{a}) = H(\boldsymbol{r}) - H(\boldsymbol{r} \mid \boldsymbol{a}).$$

Since LLM reasoning-oriented post-training primarily aims to improve answer quality in question, the generated reasoning CoT $\boldsymbol{r}$ becomes highly conditioned on the input question and is rarely optimized independently. Therefore, we propose a rational assumption as below.

**Assumption 1.** *$\pi(\boldsymbol{r})$ remains invariant during LLM RL post-training.*

Under Assumption 1, the term $H(\boldsymbol{r}) = \mathbb{E}_{\boldsymbol{r} \sim \pi_\theta}[-\log \pi_\theta(\boldsymbol{r})]$ can be treated as a constant during post-training. Moreover, leveraging the inequality $H(\boldsymbol{r} \mid \boldsymbol{a}) \leq H(\boldsymbol{r}, \boldsymbol{q} \mid \boldsymbol{a}) = H(\boldsymbol{r} \mid \boldsymbol{q}, \boldsymbol{a}) + H(\boldsymbol{q} \mid \boldsymbol{a})$, where $H(\boldsymbol{q} \mid \boldsymbol{a})$ is a constant that depends only on data distribution, we can obtain the following practical formulation.

**Theorem 1** (Surrogate IBRO objective). *Assume Assumption 1 holds, and let the reasoning trajectory be $\boldsymbol{r} = (o_1, o_2, \ldots, o_T)$. Then IBRO admits the following upper bound (up to an additive constant):*

$$\min_{\pi(\boldsymbol{r}|\boldsymbol{q})} \sum_{t=1}^{T} \left( \beta H\left(o_t \mid o_{<t}, \boldsymbol{q}, \boldsymbol{a}\right) - H\left(o_t \mid o_{<t}, \boldsymbol{q}\right) \right)$$

**Remark 2.** *(1) $H(o_t \mid o_{<t}, \boldsymbol{q}, \boldsymbol{a})$ quantifies how much knowing the correct answer $\boldsymbol{a}$ reduces uncertainty about token $o_t$. A smaller value indicates that $o_t$ is highly informative for predicting $\boldsymbol{a}$.*

```
def compute_pg_loss(log_prob, old_log_prob, advantage, clip_cfg):
    pg_loss = compute_ppo_loss(log_prob, old_log_prob, advantage,
        clip_cfg)
    entropy = compute_entropy(log_prob)
-   entropy_loss = compute_mean(entropy)
+   entropy_loss = compute_mean(entropy * advantage)
    pg_loss = pg_loss - entropy_coeff * entropy_loss
    return pg_loss
```

Listing 1: Pseudocode for computing the policy loss with IB regularization. Both `entropy` and `advantage` are token-level tensors. Only a single line is modified to incorporate IB regularization.

*Minimizing this term ensures that the reasoning chain $r$ captures key information aligned with the correct answer.*

*(2) $H(o_t \mid o_{<t}, q)$ is the standard token-level entropy. Maximizing this term promotes diversity and reduces over-reliance on the prompt $q$, thereby improving reasoning generalization.*

## 4.2 INFORMATION BOTTLENECK REGULARIZATION

While Theorem 1 provides a practical IBRO objective, computing the conditional entropy term $H(o_t \mid o_{<t}, q, a)$ requires additional inference *w.r.t.* the token distribution $p(\cdot \mid o_{<t}, q, a)$, which in turn entails extra gradient computation. In commonly adopted synchronous RL settings, the rollout generation phase accounts for more than half of the total training time (Meituan, 2025). For instance, if rollout generation consumes approximately 70% of the wall-clock time, the remaining 30% is spent on token probability and gradient computation. Ignoring device communication costs, the extra inference and gradient computation required by $H(o_t \mid o_{<t}, q, a)$ can therefore introduce an additional overhead of roughly 30%, which is substantial given the already high cost of LLM RL post-training.

To mitigate this issue, we analyze the token-level surrogate IBRO loss and derive a practical proxy. Using $I(a; o_t \mid o_{<t}, q) = H(o_t \mid o_{<t}, q) - H(o_t \mid o_{<t}, q, a)$, the token-level IBRO term can be written as

$$\ell_{\text{IB}}^t = \beta H(o_t \mid o_{<t}, q, a) - H(o_t \mid o_{<t}, q) = \left( \beta - 1 - \beta \frac{I(a; o_t \mid o_{<t}, q)}{H(o_t \mid o_{<t}, q)} \right) H_t,$$

where $H_t := H(o_t \mid o_{<t}, q)$ denotes the token-level entropy. The ratio $\frac{I(a; o_t \mid o_{<t}, q)}{H(o_t \mid o_{<t}, q)} \in [0, 1]$ is a token-wise normalized conditional mutual information that measures how strongly $o_t$ is related to the ground-truth answer $a$. On the other hand, the token advantage $A_t = A(o_t; o_{<t}, q)$ captures the token-level credit assignment. Motivated by the intuition that more valuable tokens should be more strongly correlated with the ground truth, we model the normalized mutual information as a monotone function of the advantage

$$\frac{I(a; o_t \mid o_{<t}, q)}{H(o_t \mid o_{<t}, q)} = \kappa(A_t),$$

where $\kappa$ is monotone increasing in $A_t$. By reparameterizing with $\tilde{\kappa}(A_t) := \beta\kappa(A_t) - (\beta - 1)$, we can rewrite $\ell_{\text{IB}}^t = -\tilde{\kappa}(A_t) H_t$. This leads to the following practical proxy for the IBRO objective:

$$\min \mathcal{L}_{\text{IB}} = \sum_{t=1}^{T} -\tilde{\kappa}(A_t) H_t \quad \Longleftrightarrow \quad \max \mathcal{J}_{\text{IB}} = \sum_{t=1}^{T} \tilde{\kappa}(A_t) H_t.$$

In our implementation, we instantiate $\tilde{\kappa}$ with a simple affine increasing function $\tilde{\kappa}(A_t) = \alpha A_t + \lambda$, which yields the concrete proxy

$$\max \mathcal{J}_{\text{IB}} = \sum_{t=1}^{T} (\alpha A_t + \lambda) H_t,$$

Table 1: `avg@32` scores of different regularization strategies on AMC23, AIME24, and AIME25. Boldface indicates the highest score within each RL algorithm (PPO or DAPO). `Base` reports the model performance prior to post-training.

| Method | | AMC23 | | AIME24 | | AIME25 | | Avg | |
|---|---|---|---|---|---|---|---|---|---|
| | | top@1 | top@10 | top@1 | top@10 | top@1 | top@10 | top@1 | top@10 |
| Base | | 17.3 | | 1.5 | | 1.2 | | 6.7 | |
| PPO | No reg | 63.8 | 62.8 | 17.7 | 16.8 | 13.1 | 11.9 | 31.5 | 30.5 |
| | Naive reg | 63.3 | 62.3 | 15.0 | 14.3 | 10.3 | 9.5 | 29.5 | 28.7 |
| | IB reg | **67.3** | **66.8** | **20.3** | **19.8** | **13.6** | **13.0** | **33.7** | **33.2** |
| DAPO | No reg | **86.3** | **85.7** | 18.6 | 18.2 | 17.0 | **16.3** | 40.6 | 40.1 |
| | Naive reg | 82.5 | 82.3 | 20.3 | 19.7 | 11.6 | 11.1 | 38.1 | 37.7 |
| | IB reg | 85.1 | 84.3 | **25.4** | **24.6** | **17.7** | **16.3** | **42.7** | **41.7** |

where $\alpha \in \mathbb{R}^+$ and $\lambda \in \mathbb{R}$ modulate the influence of $A_t$. Intuitively, maximizing $A_t H_t$ encourages higher entropy for critical tokens, *i.e.*, tokens with large advantages, and penalizes less informative ones. The term $\mathcal{J}_{\text{IB}}$ can be seamlessly integrated into standard RL objectives such as PPO or GRPO as an additional *IB regularization* term:

$$\max \ \mathcal{J} = \mathcal{J}_{\text{RL}} + \alpha \, \mathcal{J}_{\text{IB}},$$

where $\mathcal{J}_{\text{RL}}$ denotes the base RL objective, such as $\mathcal{J}_{\text{PPO}}$, and $\alpha > 0$ controls the regularization strength.

**Efficiency of IB Regularization**    Since both $A_t$ and $\pi_\theta(o_t \mid o_{<t}, \boldsymbol{q})$ have been obtained during RL objective computation, the proposed IB regularization is highly efficient and introduces negligible cost to the training pipeline. Moreover, the corresponding implementation is embarrassingly simple, requiring only a single line of pseudocode modification, as shown in Listing 1. A complete modification based the widely-used LLM RL training library VeRL (Sheng et al., 2025) is provided in Appendix B.

## 5 EXPERIMENTS

In this section, we evaluate the effectiveness of our IB regularization approach across various mathematical reasoning benchmarks and RL algorithms.

**Datasets and Models**    We perform RL post-training on the DAPO-Math-17K dataset (Yu et al., 2025), which consists of 17,000 mathematical questions, each paired with an integer answer. As the base model, we use Qwen2.5-7B (Yang et al., 2024) and Qwen3-14B-Base (Yang et al., 2025a), two widely adopted pre-trained LLMs that have not undergone any instruction tuning or reasoning-specific training. This makes it a suitable testbed for evaluating the effectiveness of algorithms aimed at incentivizing reasoning ability. During the post-training, we evaluate the mathematical reasoning performance of our models on seven benchmark datasets: AMC23, AIME24, , AIME25, MATH500, Olympiad, Minerva, and GSM8K.

**Setup**    We conduct LLM RL post-training based on VeRL framework (Sheng et al., 2025). To evaluate our approach, we adopt two representative RL algorithms: PPO (Schulman et al., 2017), which requires learning critic model, and DAPO (Yu et al., 2025), a GRPO variant that operates without a critic. The maximum response length is set to 20,480 tokens, and no KL regularization is applied *w.r.t.* the reference policy during training. In our PPO setup, we incorporate the `ClipHigher` strategy from DAPO (Yu et al., 2025) to mitigate entropy collapse and assess the compatibility of our IB regularization with this effective technique. The clipping parameters are set to `clip_low = 0.2` and `clip_high = 0.28`. Performance is reported using the `avg@k` metric, which measures the average pass rate over 32 sampled generations per prompt. Moreover, to comprehensively evaluate algorithm performance, we also report `pass@k`, which measures the pass rate within $k$ attempts.

**Baselines**    (1) `No reg`: vanilla RL post-training without entropy regularization; (2) `Naive reg`: RL training with standard entropy regularization, using $\alpha = 0.001$; (3) `IB reg`: RL training with

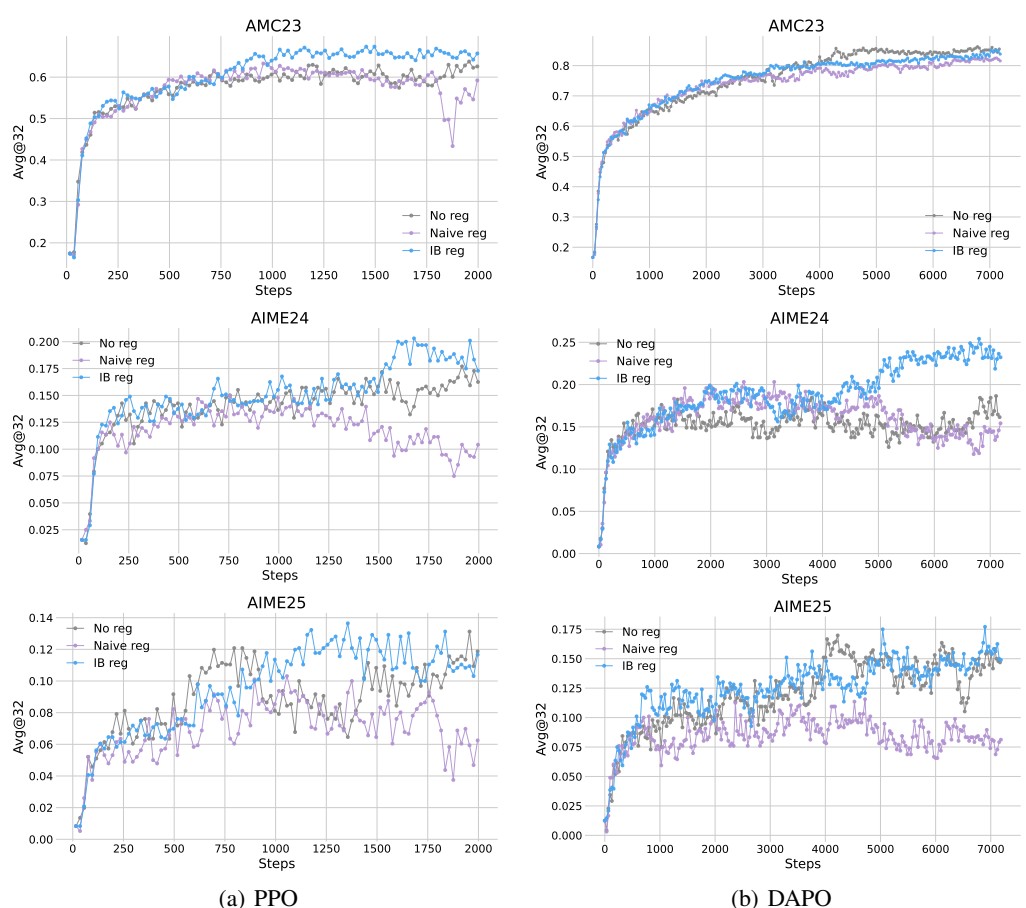

Figure 1: Plots of `avg@32` as functions of training steps in (a) PPO and (b) DAPO.

our proposed IB regularization, setting $\alpha = 0.005$ instead of $0.001$, to maintain sufficient learning signals, given the presence of both positive and negative advantages in the objective calculation.

## 5.1 MAIN RESULTS

**Qwen2.5-7B Results**  The evaluation results are presented in Table 1. To robustly assess performance improvements, we report both the best score using `top@1` and the average score across the top ten checkpoints using `top@10`. Our empirical findings reveal several key insights. First, naive entropy regularization (`Naive reg`) consistently underperforms compared to vanilla training (`No reg`), with scores dropping from 31.5 to 29.5 in PPO and from 40.6 to 38.1 in DAPO. This suggests that indiscriminate entropy injection can degrade reasoning performance. Second, our proposed IB regularization (`IB reg`) yields consistent and substantial improvements over the baseline, with average gains of two points in both PPO ($31.5 \rightarrow 33.7$) and DAPO ($40.6 \rightarrow 42.7$). These results demonstrate the effectiveness and robustness of our method.

To further support these findings, we present the training curves in Figure 1. As illustrated for PPO in Figure 1(a), the `IB reg` curves consistently outperform the baselines across all evaluation benchmarks. For DAPO, shown in Figure 1(b), although the `No reg` baseline performs best on AMC23, our `IB reg` achieves superior results on the remaining benchmarks, most notably on AIME24, where it reaches a score of 25.4, while the baselines plateau around 20.

**Qwen3-14B-Base Results**  To more comprehensively assess the effectiveness of our IB regularization, we further conduct RL post-training on the larger Qwen3-14B-Base model. Given the

Table 2: `pass@k` scores of `No reg` and `IB reg` on Qwen3-8B-Base. Boldface indicates the higher score between the two methods under the same RL algorithm (PPO or DAPO).

| Method | | Pass@k | | | | | | | | Avg |
|---|---|---|---|---|---|---|---|---|---|---|
| | | 1 | 2 | 4 | 8 | 16 | 32 | 64 | 128 | |
| AMC23 | No reg (PPO) | 81.5 | 88.9 | 92.2 | 94.0 | 95.2 | 96.4 | 97.5 | 98.7 | 93.1 |
| | IB reg (PPO) | **82.8** | **89.9** | **93.5** | **95.8** | **97.6** | **98.6** | **99.4** | **99.9** | **94.7** |
| | No reg (DAPO) | 83.7 | 90.5 | 93.8 | 95.7 | 97.1 | 97.9 | 98.5 | 99.0 | 94.5 |
| | IB reg (DAPO) | **86.9** | **92.7** | **95.1** | **96.4** | **98.5** | **98.6** | **99.6** | **100** | **95.6** |
| AIME24 | No reg (PPO) | **45.4** | 55.8 | 63.8 | 69.3 | 73.0 | 75.7 | 78.1 | 80.6 | 67.7 |
| | IB reg (PPO) | 44.7 | **56.0** | **64.7** | **71.2** | **76.1** | **79.4** | **81.1** | **82.1** | **69.4** |
| | No reg (DAPO) | 44.7 | 56.2 | **65.5** | **72.2** | **76.9** | **79.6** | 81.0 | 82.2 | 69.8 |
| | IB reg (DAPO) | **46.1** | **56.5** | 65.4 | 71.7 | 76.2 | 79.4 | **81.6** | **82.8** | **70.0** |
| AIME25 | No reg (PPO) | **34.8** | **41.6** | 47.9 | 53.2 | 57.6 | 61.5 | 64.8 | 67.4 | 53.6 |
| | IB reg (PPO) | 33.2 | 41.5 | **49.9** | **57.8** | **64.9** | **71.3** | **77.5** | **81.6** | **59.7** |
| | No reg (DAPO) | 34.2 | 41.0 | 46.9 | 52.3 | 57.1 | 62.0 | 67.4 | 73.0 | 54.2 |
| | IB reg (DAPO) | **39.3** | **46.2** | **52.5** | **58.2** | **63.5** | **67.5** | **71.0** | **73.9** | **59.0** |

Table 3: `pass@1` scores of `No reg` and `IB reg` on Qwen3-8B-Base. Boldface indicates the highest score within each RL algorithm (PPO or DAPO).

| | Method | AMC23 | AIME24 | AIME25 | MATH500 | Olympiad | Minerva | GSM8K | Avg |
|---|---|---|---|---|---|---|---|---|---|
| | Base | 22.2 | 3.3 | 2.4 | 68.8 | 36.4 | 23.5 | 41.5 | 28.3 |
| PPO | No reg | 81.5 | **45.4** | **34.8** | 86.0 | 55.9 | 32.4 | 54.9 | 55.8 |
| | IB reg | **82.8** | 44.7 | 33.2 | **88.0** | **58.9** | **34.6** | **67.8** | **58.6** |
| DAPO | No reg | 83.7 | 44.7 | 34.2 | 86.4 | 58.0 | **37.5** | 64.5 | 58.4 |
| | IB reg | **86.9** | **46.1** | **39.3** | **88.2** | **58.5** | 33.8 | **69.3** | **60.3** |

consistently weak performance of `Naive reg`, we focus our comparison on `IB reg` versus the `No reg` baseline. All models are trained for 800 steps, and the final checkpoint is used for evaluation.

We report `pass@k` scores on AMC23, AIME24, and AIME25 in Table 2. As shown, `IB reg` yields stable and consistent improvements across all values of $k$. Moreover, Table 3 presents `pass@1` scores on a broader set of mathematical reasoning tasks. Our method again demonstrates consistent gains over the baseline, achieving an average improvement of approximately two points for both PPO ($55.8 \rightarrow 58.6$) and DAPO ($58.4 \rightarrow 60.3$).

## 5.2 EMPIRICAL ANALYSIS

We also conduct a detailed analysis on Qwen2.5-7B by comparing different strategies in terms of entropy dynamics and response length.

**Entropy Dynamics** Beyond performance metrics, we examine the entropy dynamics during post-training under both PPO and DAPO, as illustrated in Figure 2. Compared to the no-regularization baseline, naive entropy regularization partially mitigates entropy collapse. However, it frequently drives the entropy to excessively high levels in the later stages of training, sometimes even exceeding its initial value. In contrast, IB regularization maintains entropy at a similar magnitude to the vanilla baseline, while exhibiting more stable and controlled behavior throughout training.

These observations offer several insights. First, given the inferior empirical performance of naive entropy regularization, excessive entropy can be as detrimental as entropy collapse for LLM reasoning, leading to unfocused exploration. Second, IB regularization does not seek to increase token entropy uniformly. Instead, it selectively redistributes entropy by encouraging higher entropy for critical tokens that benefit from exploration, while reducing entropy for less informative tokens to maintain coherence and fluency. Crucially, as IB regularization preserves the overall entropy scale, it is highly compatible with existing training pipelines: in practice, modulating the degree of off-policyness

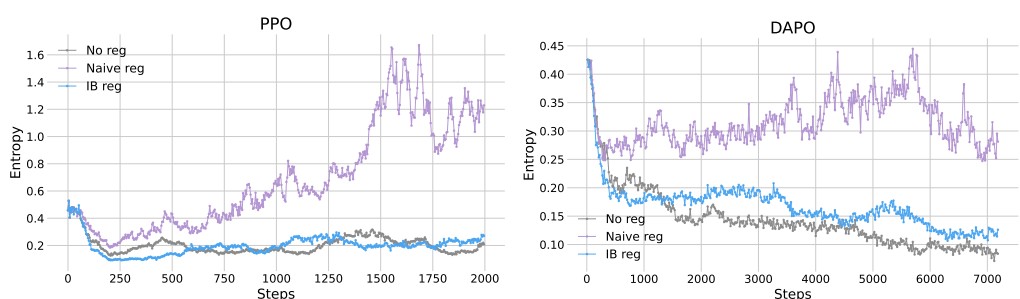

Figure 2: Plots of entropy as functions of training steps.

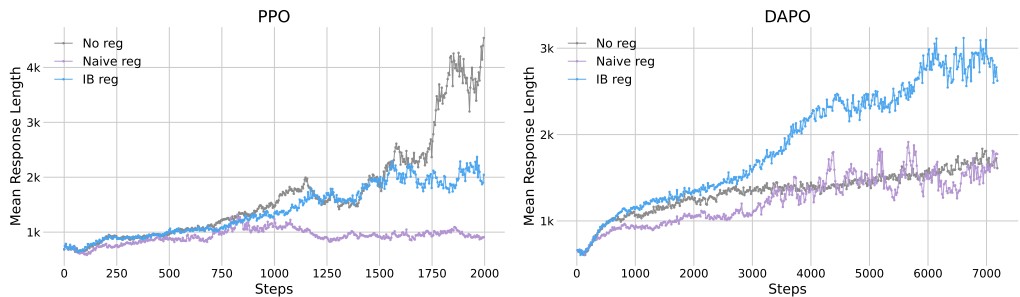

Figure 3: Plots of mean response length as functions of training steps.

in RL, tuning sampling temperature, or employing the `ClipHigher` strategy, provides a more stable and general mechanism for maintaining a balanced entropy range, compared to explicit entropy regularization (He et al., 2025; Yu et al., 2025; Liu et al., 2025a). As a result, our IB regularization can be seamlessly integrated into well-tuned setups without significantly disrupting the entropy dynamics.

**Response Length Analysis**  Previous studies have shown that improvements in response length are often closely associated with gains in reasoning accuracy, as longer responses tend to reflect more complete and detailed reasoning processes. Figure 3 illustrates the evolution of mean response length throughout post-training. We observe two key patterns: (1) IB regularization does not consistently produce longer responses compared to the vanilla baseline. Concretely, it yields shorter responses under PPO and longer ones under DAPO. Nevertheless, the response length under IB regularization exhibits stable growth and consistently remains within a desirable range of $2K-3K$ tokens; and (2) naive entropy regularization tends to shorten responses, especially under PPO, with a less pronounced effect in DAPO. This outcome, while somewhat counter-intuitive, has not been formally documented in prior work. It challenges the common belief that higher entropy promotes greater exploration and thus results in longer reasoning trajectories.

We attribute this phenomenon to the behavior of the end-of-sequence token (`[EOS]`). During early decoding stages, the probability of emitting `[EOS]` is typically low. However, naive entropy regularization increases entropy uniformly across all tokens, which flattens the output probability distribution and can inadvertently raise the likelihood of generating `[EOS]`, thereby causing premature truncation and shorter responses. In contrast, IB regularization selectively increases entropy for critical tokens while suppressing it for less informative ones. This suppression often raises the relative probability of sampled uninformative tokens while reducing that of others, such as `[EOS]`. As a result, IB regularization mitigates premature termination and better preserves the response length required for effective multi-step reasoning.

## 6   DISCUSSION AND LIMITATION

**Discussion**  The proposed IB regularization can be interpreted as a token-level entropy regularization scheme, where the regularization strength for each token is weighted by its corresponding advantage.

In PPO, token-level advantages vary within a single response and are provided by the critic model. As a result, critical tokens, *i.e.*, tokens with higher advantages, receive stronger entropy regularization. This mechanism focuses optimization on informative positions, promotes targeted exploration, and contributes to improved reasoning generalization.

In contrast, GRPO and DAPO assign the same scalar advantage to all tokens within a response, based on the group-normalized final reward. Under this setting, IB regularization degrades to applying a positive entropy regularization to correct responses and a negative one to incorrect responses. Since updates on correct responses tend to concentrate the output distribution and reduce entropy, while updates on incorrect responses often flatten the distribution and increase entropy, IB regularization serves as a soft constraint or entropy-aware damping force that counteracts excessive entropy reduction on correct responses and curbs entropy explosion on incorrect ones. This mechanism encourages conservative token-level entropy adjustments, helping to maintain a well-balanced entropy profile and ensuring stable training.

**Limitation**   While our results demonstrate the effectiveness of IB regularization, several limitations remain. First, the regularization strength coefficient requires careful tuning to achieve optimal performance, and its optimal value may vary across tasks, model sizes, and training stages. Developing automated or adaptive tuning strategies remains an open challenge. Second, due to computational constraints, our experiments are conducted on models with 7B and 14B parameters. The scalability and effectiveness of IB regularization on larger LLMs, such as those with over 100B parameters, have not yet been validated. Exploring this setting is non-trivial, as running RL post-training on very large LLMs demands substantially greater computational resources, potentially exceeding an order of magnitude beyond our current setup. We leave this investigation to future work.

## 7    CONCLUSION

In this paper, we introduce an information-theoretic framework called *information bottleneck-aware reasoning optimization* (IBRO) to optimize reasoning trajectories in large language models (LLMs). Grounded in information bottleneck principle, IBRO formalizes reasoning effectiveness by encouraging trajectories to be informative regarding the correct answer while remaining generalizable across different prompts. We further derived a token-level surrogate objective and proposed a practical approximation, termed *IB regularization*, which modulates token-level entropy according to token-level advantage readily available from RL frameworks. The method requires no additional computation and can be implemented with only one line of code modification. Empirical evaluations across several mathematical reasoning benchmarks demonstrate that integrating IB regularization into existing RL algorithms consistently enhances reasoning accuracy and stability. Our results underline the significance of information-theoretic insights in optimizing LLM reasoning, providing theoretical foundations and practical tools for future research in this direction.

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

## A  PROOF OF THEOREM 1

Our proof is based on Assumption 1, which states that the marginal entropy $H(r)$ remains approximately constant during training. We empirically verify this assumption in Appendix F. We also use the inequality $H(r \mid a) \leq H(r, q \mid a) = H(r \mid q, a) + H(q \mid a)$, where $H(q \mid a)$ depends only on the dataset distribution and is therefore constant.

*Proof.* The IBRO objective seeks to minimize

$$\mathcal{J}_{\text{IBRO}} = I(q; r) - \beta I(r; a).$$

Expressing the mutual information terms using entropy yields

$$\mathcal{J}_{\text{IBRO}} = H(r) - H(r \mid q) - \beta\big(H(r) - H(r \mid a)\big)$$
$$= \beta H(r \mid a) - H(r \mid q) + (1 - \beta)\,H(r).$$

Applying the inequality above gives

$$\mathcal{J}_{\text{IBRO}} \leq \beta H(r \mid q, a) - H(r \mid q) + \beta H(q \mid a) + (1 - \beta)H(r).$$

Both $H(q \mid a)$ and $H(r)$ are constants during LLM RL post-training. Under Assumption 1, minimizing $\mathcal{J}_{\text{IBRO}}$ is therefore equivalent (up to additive constants) to minimizing

$$\beta H(r \mid q, a) - H(r \mid q).$$

With $r = (o_1, o_2, \ldots, o_T)$, the chain rule of entropy decomposes this expression into token-level terms, yielding the desired surrogate objective. $\square$

## B  IB REGULARIZATION MODIFICATION ON VERL

We provide an example of modifying the VeRL code to switch from naive entropy regularization to our proposed IB regularization; please refer to Listing 2.

```
    if entropy_coeff != 0:
+       token_advantages = data["advantages"]
-       entropy_loss = agg_loss(loss_mat=entropy, loss_mask=
    response_mask, loss_agg_mode=loss_agg_mode)
+       entropy_loss = agg_loss(loss_mat=entropy * token_advantages,
     loss_mask=response_mask, loss_agg_mode=loss_agg_mode)
        # compute policy loss
        policy_loss = pg_loss - entropy_loss * entropy_coeff
    else:
        policy_loss = pg_loss
```

Listing 2: IB regulareization modification on VeRL.

## C  IMPLEMENTATION DETAILS

The pre-trained LLM of Qwen2.5-7B can be download via `https://huggingface.co/Qwen/Qwen2.5-7B`. The training dataset DAPO-Math-17K is available at `https://huggingface.co/datasets/BytedTsinghua-SIA/DAPO-Math-17k`, and the evaluation datasets of AMC23, AIME24, and AIME25 can be download on `https://huggingface.co/math-ai`. Table 4 lists the key hyperparameters used in PPO and DAPO. All experiments are conducted on $4 \times 8$ NVIDIA H20 GPUs.

Table 4: Key hyperparameters for PPO and DAPO. "—" denotes not used.

| Category | PPO | DAPO |
|---|---|---|
| *Sampling and Validation* | | |
| Temperature | 1.0 | 1.0 |
| Top-$p$ / Val Top-$p$ | 1.0 / 0.7 | 1.0 / 0.7 |
| *Clipping* | | |
| Clip ratio (low / high) | 0.2 / 0.28 | 0.2 / 0.28 |
| *Sequence Limits* | | |
| Max prompt / response length | 2048 / 20480 | 2048 / 20480 |
| Overlong buffer (on/off) | off | on |
| Buffer length / penalty | — | 4096 / 1.0 |
| *Batching* | | |
| Train batch / mini-batch size | 1024 / 256 | 512 / 32 |
| Gen batch size | — | $512 \times 3$ |
| Responses per prompt | — | 16 |
| *Optimization* | | |
| Loss aggregation | `token-mean` | `token-mean` |
| Actor LR | $1e-6$ | $1e-6$ |
| Critic LR | $1e-5$ | — |
| LR warmup steps | 10 | 10 |
| Critic warmup steps | 5 | — |

## D  THE USE OF LLMS

We utilized LLMs solely for polishing the writing, with the aim of improving linguistic quality and ensuring a more formal academic style.

## E  COMPARISON ON COMPUTATIONAL OVERHEAD

We compare the computational overhead of our `IB reg` with the `No reg` baseline on Qwen3-14B-Base; see Table 5. TFLOPs is reported as the average per training step over 800 steps, and the wall-clock time reflects the total end-to-end training duration. All experiments are conducted on 32 NVIDIA H20 GPUs. As shown in the table, `IB reg` and `No reg` exhibit similar computational overhead, supporting our claim that the proposed approach introduces negligible additional cost.

Table 5: Computational overhead on Qwen3-14B-Base. TFLOPs is reported as the average over the entire training period, and wall-clock time denotes the total end-to-end training duration.

| | Method | TFLOPs | Wall-Clock Time (hours) |
|---|---|---|---|
| PPO | No reg | 61.3 | 75.1 |
| PPO | IB reg | 62.3 | 71.6 |
| DAPO | No reg | 99.4 | 31.3 |
| DAPO | IB reg | 100.7 | 32.5 |

## F  ENTROPY EXPERIMENTS

In the RLVR setting, the model optimizes the conditional distribution $\pi(r|q)$ given a training prompt $q$. Thus, the entropy relevant to RLVR is the conditional entropy $H(r \mid q)$, whereas Assumption 1 concerns the stability of the marginal, or vanilla, entropy $H(r)H(r)$ throughout training. To examine this assumption, we use the DAPO-Math-17k-Qwen3-235B-A22B-Thinking CoT dataset (Zhou,

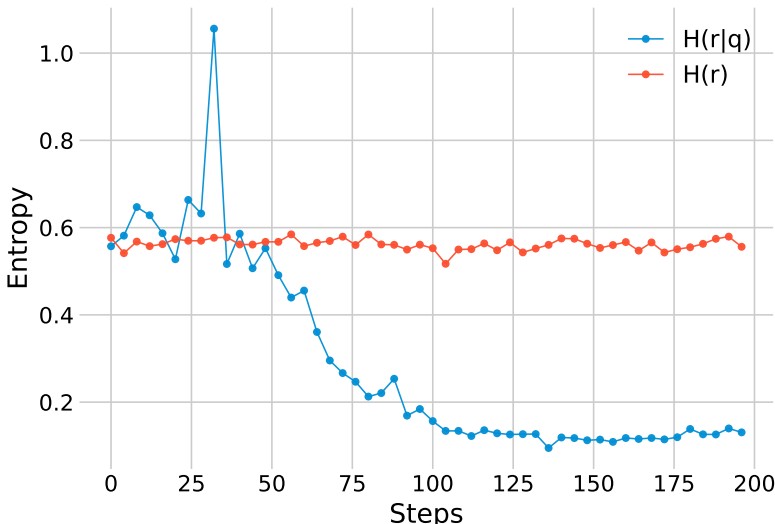

Figure 4: Entropy $H(r)$ and conditional entropy $H(r \mid q)$ trajectories over the course of training.

2025) as $r$, and analyze both $H(r)$ and $H(r \mid q)$ during RL training on Qwen2.5-7B with DAPO. The trajectories of the two entropies are shown in Figure 4. As illustrated, $H(r \mid q)$ decreases rapidly once training begins, while $H(r)$ remains stable, thereby supporting Assumption 1.

## G MORE THEORETICAL ANALYSIS

With the IBRO framework, we further derive a generalization bound based on information-theoretic analysis in (Kawaguchi et al., 2023), and the proof detail can be found in Appendix G.1.

**Theorem 2** (IBRO generalization bound). *Let $\pi$ be a LLM, training dataset $\mathcal{S} = \{(q_i, a_i)\}_{i=1}^m$ are i.i.d. drawn from the joint data distribution $\mathcal{D}$. Suppose the LLM parameters are updated by $\Delta\theta$ during learning. Let $\mathrm{ACC}(\mathcal{S})$ and $\mathrm{ACC}(\mathcal{D})$ denote the empirical and population accuracy of $\pi$ after training, respectively. Define $\mathcal{L}_{IB} = \beta I(r \mid q, a) - I(r \mid q)$ computed w.r.t. $\mathcal{D}$. If $\beta \geq 2$, then, for any $\delta > 0$, with probability at least $1 - \delta$ over the sampling of $\mathcal{S}$, the generalization gap $\Delta(\mathcal{S}) = |\mathrm{ACC}(\mathcal{S}) - \mathrm{ACC}(\mathcal{D})|$ satisfies*

$$\Delta(\mathcal{S}) \lesssim \sqrt{\frac{\mathcal{L}_{IB} + \|\Delta\theta\|^2 + \log\frac{1}{\delta}}{m}} + \tilde{\mathcal{O}}\left(\sqrt{\frac{\|\Delta\theta\|^2 + 1}{m}}\right).$$

Compared to supervised fine-tuning (SFT), RL-based LLM training typically yields highly sparse parameter updates (Mukherjee et al., 2025) and small KL divergence w.r.t. the initial parameters (Rajani et al., 2025), resulting in a small $\|\Delta\theta\|$. This implies that the generalization bound is predominantly governed by $\mathcal{L}_{IB}$ during LLM RL training.

### G.1 PROOF OF THEOREM 2

Our proof is based on the theoretical results in (Kawaguchi et al., 2023), which we first outline a simply version as below.

**Notation** We denote the input and output variables by $X$ and $Y$, respectively. Consider a neural network $f = g \circ \phi$, which comprises two components: an encoder $\phi$ that maps inputs $X$ to latent features $Z = \phi(X)$, and a predictor $g$ that generates predictions $g(Z)$ based on these latent representations. Let $\mathcal{S} = \{(x_i, y_i)\}_{i=1}^m$ be a training dataset consisting of $m$ examples drawn independently and identically distributed (i.i.d.) from a joint distribution $\mathcal{P}$ over $\mathcal{X} \times \mathcal{Y}$, with $x_i \in \mathcal{X}$ and $y_i \in \mathcal{Y}$. Given a bounded loss function $\ell : \mathcal{X} \times \mathcal{Y} \to \mathbb{R}^+$, the generalization gap, defined as the

difference between the expected and empirical losses, is expressed as:

$$\Delta(\mathcal{S}) := \mathbb{E}_{(X,Y)\sim\mathcal{P}} \left[\ell(f^{\mathcal{S}}(X), Y)\right] - \frac{1}{m}\sum_{i=1}^{m} \ell(f^{\mathcal{S}}(\boldsymbol{x}_i), \boldsymbol{y}_i),$$

where $f^{\mathcal{S}}$ denotes the neural network trained on the dataset $\mathcal{S}$.

**Theorem 3** (Theorem 2 in (Kawaguchi et al., 2023)). *Given a network $f = g \circ \phi$ trained on the dataset $\mathcal{S}$, the dataset size $|\mathcal{S}| = m$. Then, for any $\delta > 0$, with probability at least $1 - \delta$ over the training set $\mathcal{S}$, the following generalization bound holds:*

$$\Delta(\mathcal{S}) \lesssim \sqrt{\frac{I(X;Z|Y) + I(\phi;S) + H(Z|X,Y) + \log\frac{1}{\delta}}{m}} + \tilde{\mathcal{O}}\left(\sqrt{\frac{I(\phi, S) + 1}{m}}\right) \quad (1)$$

To upper bound the mutual information between model parameters $\phi$ and dataset $S$, we invoke the following lemma.

**Lemma 1.** *Let the encoder parameters $\theta \in \mathbb{R}^d$ be updated from initialization $\theta_0$ to $\theta = \theta_0 + \Delta\theta$ after training on dataset $S$. Let the prior distribution be $P_0 = \mathcal{N}(\theta_0, \sigma^2 I)$, and the posterior $P_{\Theta|S} = \delta(\theta_0 + \Delta\theta)$. Then, for any $\sigma > 0$, the mutual information satisfies:*

$$I(\theta; S) \leq \frac{\|\Delta\theta\|^2}{2\sigma^2} + \frac{d}{2}\log(2\pi\sigma^2).$$

*Proof.* By definition,

$$I(\theta; S) = \mathbb{E}_S\left[\text{KL}(P_{\Theta|S}\|P_\Theta)\right] \leq \mathbb{E}_S\left[\text{KL}(P_{\Theta|S}\|P_0)\right],$$

using convexity of KL and data-independent prior $P_0$. Since $P_{\Theta|S} = \delta(\theta_0 + \Delta\theta)$ and $P_0 = \mathcal{N}(\theta_0, \sigma^2 I)$, we compute

$$\text{KL}(\delta(\theta_0 + \Delta\theta) \,\|\, \mathcal{N}(\theta_0, \sigma^2 I)) = \frac{\|\Delta\theta\|^2}{2\sigma^2} + \frac{d}{2}\log(2\pi\sigma^2).$$

$\square$

As $H(\boldsymbol{r} \mid \boldsymbol{q}, \boldsymbol{a}) = H(\boldsymbol{r} \mid \boldsymbol{q})$ under the Markov chain $\boldsymbol{a} \leftrightarrow \boldsymbol{q} \leftrightarrow \boldsymbol{r}$, we have

$$I(\boldsymbol{q}; \boldsymbol{r} \mid \boldsymbol{a}) = H(\boldsymbol{r} \mid \boldsymbol{a}) - H(\boldsymbol{r} \mid \boldsymbol{q}, \boldsymbol{a}) = H(\boldsymbol{r} \mid \boldsymbol{a}) - H(\boldsymbol{r} \mid \boldsymbol{q}).$$

Then

$$H(\boldsymbol{r} \mid \boldsymbol{a}) \leq H(\boldsymbol{r}, \boldsymbol{q} \mid \boldsymbol{a}) = H(\boldsymbol{r} \mid \boldsymbol{q}, \boldsymbol{a}) + H(\boldsymbol{q} \mid \boldsymbol{a}),$$

which gives the inequality

$$I(\boldsymbol{q}; \boldsymbol{r} \mid \boldsymbol{a}) \leq H(\boldsymbol{r} \mid \boldsymbol{q}, \boldsymbol{a}) + H(\boldsymbol{q} \mid \boldsymbol{a}) - H(\boldsymbol{r} \mid \boldsymbol{q}).$$

Recall the surrogate IBRO loss

$$\mathcal{L}_{\text{IB}} = \beta\, H(\boldsymbol{r} \mid \boldsymbol{q}, \boldsymbol{a}) - H(\boldsymbol{r} \mid \boldsymbol{q}),$$

then for $\beta \geq 2$, we obtain

$$I(\boldsymbol{q}; \boldsymbol{r} \mid \boldsymbol{a}) + H(\boldsymbol{r} \mid \boldsymbol{q}, \boldsymbol{a}) \leq 2H(\boldsymbol{r} \mid \boldsymbol{q}, \boldsymbol{a}) + H(\boldsymbol{q} \mid \boldsymbol{a}) - H(\boldsymbol{r} \mid \boldsymbol{q}) \leq \mathcal{L}_{\text{IB}} + H(\boldsymbol{q} \mid \boldsymbol{a}).$$

Therefore, under the mild condition $\beta \geq 2$, the quantity $\mathcal{L}_{\text{IB}} + H(\boldsymbol{q} \mid \boldsymbol{a})$ serves as an upper bound on $I(\boldsymbol{q}; \boldsymbol{r} \mid \boldsymbol{a}) + H(\boldsymbol{r} \mid \boldsymbol{q}, \boldsymbol{a})$. Since $H(\boldsymbol{q} \mid \boldsymbol{a})$ is a constant that depends only on the data distribution $\mathcal{P}$, combining Theorem 3 and Lemma 1 yields the desired bound.

