# OpenReview forum: "Revisiting LLM Reasoning via Information Bottleneck"
_ICLR.cc/2026/Conference — Submitted to ICLR 2026_

### Official Review · Reviewer_qmnN · 2025-10-24

**Soundness:** 3
**Presentation:** 3
**Contribution:** 3
**Rating:** 6
**Confidence:** 4

**Summary:**

This paper presents a theoretical characterization of large language model (LLM) reasoning grounded in the Information Bottleneck principle, introducing IB-aware Reasoning Optimization (IBRO), a framework that encourages reasoning trajectories to be both informative about the correct answer and generalizable across diverse prompts. This technique integrates seamlessly into existing RL-based post-training frameworks with negligible computational overhead. The approach demonstrates consistent performance improvements across multiple benchmarks.

**Strengths:**

- This paper establishes a theoretical framework for LLM reasoning, grounded in the Information Bottleneck principle.

- The framework is designed to maximize the informativeness of reasoning trajectories regarding the final answer while ensuring they remain generalizable across diverse prompts.

**Weaknesses:**

- It is recommended that the notation in Theorem 2 be carefully reviewed. The current use of $I(r|q,a)$  in the definition of $\mathcal{L}_IB$ is inconsistent with the paper's own context, where conditional entropy $H(r|q,a)$ is intended. Correcting this is essential for the theoretical rigor of the presentation.
- There are also some works that apply information theory to LLM [A-D]. It is recommended to supplement the relevant discussions and explain the differences with them.

[A] Understanding chain-of-thought in llms through information theory. [B] Learning to think: Information-theoretic reinforcement fine-tuning for llms. [C] EVINCE: Optimizing Multi-LLM Dialogues Using Conditional Statistics and Information Theory. [D] Exploring Information Processing in Large Language Models: Insights from Information Bottleneck Theory.

- The approximation $\lambda_t \approx -A_t$ is a critical yet under-justified step. The authors should better explain why the negative advantage can replace the ground-truth-dependent $\lambda_t$. While intuitively plausible (if a token is important, its uncertainty when knowing the answer should be low), this claim lacks both theoretical justification and empirical support (e.g., showing negative correlation between $A_t$ and $H(o_t | ..., a)$). We recommend adding a paragraph to clarify this approximation's rationale or providing a small validation experiment.

**Questions:**

- The paper seems to implicitly assume that the reasoning process is a monotonic progression toward the correct answer. However, in practice, reasoning can be non-monotonic, involving exploration, temporary detours, or even incorrect sub-goals. Could the authors clarify the scope of their framework in handling such non-ideal, yet common, reasoning trajectories? A discussion on how the token-level advantage $A_t$ captures and manages these dynamics would strengthen the theoretical grounding of the work.

- The authors could discuss whether their method enhances interpretability. For example, examining the alignment between high-advantage tokens and key reasoning steps would be valuable.
- Please supplement the relevant discussions about the differences and connection with prior  information theory-based works.

- Please evaluate the performance of the proposed method against a broader range of existing entropy regularization strategies to better validate its advantages.

- The authors emphasize the "negligible" overhead of IB regularization but provide no specific profiling data. Given the substantial cost of large-scale RL training, could the authors present the actual wall-clock time or FLOPs overhead per training step (or throughout training) introduced by IB regularization, quantified as a percentage compared to the no-regulation baseline? This would provide a critical data point for practitioners considering the adoption of this method.

---

> ### Author Response · Authors · 2025-11-26
> **To Reviewer qmnN (1/2)**
>
> **Q1:** *It is recommended that the notation in Theorem 2 be carefully reviewed.*
>
> **A1:** Thanks and addressed. We have moved Theorem 2 to Appendix to in the revised version to keep the main presentation focused on our approach.
>
> &nbsp;
>
> **Q2:** *There are also some works that apply information theory to LLM. It is recommended to supplement the relevant discussions and explain the differences with them.*
>
> **Q2:** Thank you for the suggestion. We have accordingly revised it; please refer to Lines 103-107.
>
> &nbsp;
>
> **Q3:** *The approximation is a critical yet under-justified step. The authors should better explain why the negative advantage can replace the ground-truth-dependent . While intuitively plausible (if a token is important, its uncertainty when knowing the answer should be low), this claim lacks both theoretical justification and empirical support (e.g., showing negative correlation between and ). We recommend adding a paragraph to clarify this approximation's rationale or providing a small validation experiment.*
>
> **A3:** Thanks for your suggestion. We have revised the paper and added further explanation on using advantage as an intuitive proxy for the IBRO objective. Concretely, we reformulize the IBRO objective as
>
> $$
> \ell_{\text{IB}}^{t}
> = \beta H(o_t \mid o_{<t}, \boldsymbol{q}, \boldsymbol{a}) - H(o_t \mid o_{<t}, \boldsymbol{q})
> = \left(\beta - 1 - \beta  \frac{ I(\boldsymbol{a} ; o_t \mid o_{<t}, \boldsymbol{q}) }
>       { H(o_t \mid o_{<t}, \boldsymbol{q}) }\right) H_t
> $$
>
> The ratio $\frac{I\left(\boldsymbol{a} ; o_t \mid o_{<t}, \boldsymbol{q}\right)}{
>   H\left(o_t \mid o_{<t}, \boldsymbol{q}\right)
> } \in [0, 1]$ is a token-wise normalized conditional mutual information that measures how strongly \(o_t\) is related to the ground-truth answer \(\boldsymbol{a}\). On the other hand, the token advantage $A_t = A(o_t ; o_{<t}, \boldsymbol{q})$ captures the token-level credit assignment. Motivated by the intuition that more valuable tokens should be more strongly correlated with the ground truth, we model the normalized mutual information as a monotone function of the advantage
> $$
> \frac{I\left(\boldsymbol{a} ; o_t \mid o_{<t}, \boldsymbol{q}\right)}{H\left(o_t \mid o_{<t}, \boldsymbol{q}\right)}
> = \kappa(A_t),
> $$
> where $\kappa$ is monotone increasing in $A_t$. By reparameterizing with $\tilde{\kappa}(A_t) := \beta \kappa(A_t) - ({\beta - 1})$, we can rewrite $\ell_\texttt{IB}^t
> = -\tilde{\kappa}(A_t)\, H_t$. This leads to the following practical proxy for the IBRO objective:
> $$
> \min \ \mathcal{L}_{\texttt{IB}} = \sum_t^T -  \tilde{\kappa}(A_t) H_t \quad\Longleftrightarrow\quad \max \ \mathcal{J} = \sum_t^T \tilde{\kappa}(A_t)\ H_t
> $$
> In our implementation, we instantiate $\tilde{\kappa}$ with a simple affine increasing function $\tilde{\kappa}(A_t) =  \alpha A_t + \lambda $, which yields the desired proxy with advantages. More expressive forms of $\tilde{\kappa}$ can also be used, and exploring such variants is an interesting direction for future work.
>
> We have updated the paper accordingly, and the detail can be found in Lines 246 - 285 of the revised version.
>
> &nbsp;
>
> **Q4:** *The paper seems to implicitly assume that the reasoning process is a monotonic progression toward the correct answer. However, in practice, reasoning can be non-monotonic, involving exploration, temporary detours, or even incorrect sub-goals. Could the authors clarify the scope of their framework in handling such non-ideal, yet common, reasoning trajectories? A discussion on how the token-level advantage captures and manages these dynamics would strengthen the theoretical grounding of the work.*
>
> **A4:** Thank you for the insightful comment. We fully agree that real-world reasoning is often non-monotonic, involving exploration, temporary detours, and even incorrect intermediate steps. While a detailed analysis of these dynamics is beyond the scope of this paper, investigating how LLM reasoning evolves and transitions across different states is an interesting and important direction for future work.
>
> &nbsp;
>
> **Q5:** *The authors could discuss whether their method enhances interpretability. For example, examining the alignment between high-advantage tokens and key reasoning steps would be valuable.*
>
> A5: Thanks for your comments. Our IB regularization applies to both PPO-style algorithms, where token advantages differ across positions within a sequence, and GRPO-style algorithms, where all tokens in a sequence share the same estimated advantage. Because of this, identifying key reasoning steps purely through token-level advantage is not consistently applicable across algorithms.
>
> &nbsp;
>
> **Q6:** *Please supplement the relevant discussions about the differences and connection with prior information theory-based works.*
>
> **A6:** Please refer to **A2**.

---

> ### Author Response · Authors · 2025-11-26
> **To Reviewer qmnN (2/2)**
>
> **Q7:** *Please evaluate the performance of the proposed method against a broader range of existing entropy regularization strategies to better validate its advantages.*
>
> **A7:**  Thanks for your suggestion. Since LLM RLVR is still an emerging research area, there are currently very few widely adopted entropy-regularization baselines. For this reason, we included the most direct variant, namely naive entropy regularization (Naive reg). We will continue to monitor developments in this space and plan to compare against a broader set of entropy-regularization strategies in future work.
>
> &nbsp;
>
> **Q8:** *The authors emphasize the "negligible" overhead of IB regularization but provide no specific profiling data. Given the substantial cost of large-scale RL training, could the authors present the actual wall-clock time or FLOPs overhead per training step (or throughout training) introduced by IB regularization, quantified as a percentage compared to the no-regulation baseline? This would provide a critical data point for practitioners considering the adoption of this method.*
>
> **A8:** Thank you for the suggestion. Since both the advantages and token probabilities required by IB regularization are already available in standard LLM RL pipelines, our method introduces negligible additional computational cost compared with the No-reg baseline. To empirically confirm this, we report the FLOPs and training wall-clock time in the table below. TFLOPs is measured as the average per training step over 800 steps, and the wall-clock time reflects the total end-to-end training duration. All experiments are conducted on 32 NVIDIA H20 GPUs. As shown, IB reg and the No-reg baseline exhibit nearly identical computational overhead, supporting our claim that the additional cost is negligible. We have updated the paper accordingly, and further details can be found in Table 5 of the revised version.
>
> - Computational overhead of Qwen3-14B-Base on PPO
> | Method | TFLOPs | Wall-Clock Time (hours) |
> | :----: | :----: | :---------------------: |
> | No reg |  61.3  |          75.1           |
> | IB reg |  62.3  |          71.6           |
>
> - Computational overhead of Qwen3-14B-Base on DAPO
> | Method | TFLOPs | Wall-Clock Time (hours) |
> | :----: | :----: | :---------------------: |
> | No reg |  99.4  |          31.3           |
> | IB reg | 100.7  |          32.5           |

---

### Official Review · Reviewer_baqE · 2025-10-31

**Soundness:** 3
**Presentation:** 3
**Contribution:** 2
**Rating:** 4
**Confidence:** 3

**Summary:**

The paper introduces Information Bottleneck–aware Reasoning Optimization (IBRO), an information-theoretic framework for improving LLM reasoning. It derives a token-level surrogate objective and proposes IB regularization, which adjusts token entropy based on token advantages. The method integrates easily into PPO and DAPO with minimal code change and yields small but consistent improvements on AMC23, AIME24, and AIME25 benchmarks, along with more stable entropy dynamics.

**Strengths:**

- IB regularization is elegant, lightweight for Reasoning RL
- Offers a clear theoretical perspective on LLM reasoning through the information bottleneck
- Includes a generalization bound that connects theory and empirical outcomes

**Weaknesses:**

- The improvements are relatively small (around +2 points) and may not be statistically significant. Figure 1 also suggests that the choice of training step or checkpoint may play a more substantial role in performance variation than the proposed regularization
- Experiments limited to one model (Qwen2.5-7B) and math reasoning tasks

**Questions:**

- How well does IB regularization generalize to non-math reasoning tasks?
- Hot to empirically confirm that token advantage aligns with informational importance?
- How robust is the method to different β and α settings?
- What are the expected effects when scaling to larger LLMs? Scaling?

---

> ### Author Response · Authors · 2025-11-26
> **To Reviewer baqE (1/2)**
>
> **Q1:** *Experiments limited to one model (Qwen2.5-7B) and math reasoning tasks*
>
> **A1:** Thank you for the suggestion. During the rebuttal period, we conducted additional experiments covering more models, evaluation tasks, and metrics. In particular, we performed RL post-training on the Qwen3-14B-Base model using both PPO and DAPO. The pass@k results on AMC23, AIME24, and AIME25, comparing our IB regularization with the No-reg baseline, are presented below. As shown, IB reg delivers stable and consistent improvements across all values of k.
>
> - Pass@k scores of Qwen3-14B-Base trained with PPO on AMC23:
> | pass@k | 1        | 2        | 4        | 8        | 16       | 32       | 64       | 128      | Avg      |
> | :----: | -------- | -------- | -------- | -------- | -------- | -------- | -------- | -------- | -------- |
> | No reg | 81.5     | 88.9     | 92.2     | 94.0     | 95.2     | 96.4     | 97.5     | 98.7     | 93.1     |
> | IB reg | **82.8** | **89.9** | **93.5** | **95.8** | **97.6** | **98.6** | **99.4** | **99.9** | **94.7** |
>
> - Pass@k scores of Qwen3-14B-Base trained with DAPO on AMC23:
> | pass@k |    1     |    2     |    4     |    8     |    16    |    32    |    64    |   128   |   Avg    |
> | :----: | :------: | :------: | :------: | :------: | :------: | :------: | :------: | :-----: | :------: |
> | No reg |   83.7   |   90.5   |   93.8   |   95.7   |   97.1   |   97.9   |   98.5   |  99.0   |   94.5   |
> | IB reg | **86.9** | **92.7** | **95.1** | **96.4** | **98.5** | **98.6** | **99.6** | **100** | **95.6** |
>
> - Pass@k scores of Qwen3-14B-Base trained with PPO on AIME24:
> | pass@k |    1     |    2     |    4     |    8     |    16    |    32    |    64    |   128    |   Avg    |
> | :----: | :------: | :------: | :------: | :------: | :------: | :------: | :------: | :------: | :------: |
> | No reg | **45.4** |   55.8   |   63.8   |   69.3   |   73.0   |   75.7   |   78.1   |   80.6   |   67.7   |
> | IB reg |   44.7   | **56.0** | **64.7** | **71.2** | **76.1** | **79.4** | **81.1** | **82.1** | **69.4** |
>
> - Pass@k scores of Qwen3-14B-Base trained with DAPO on AIME24:
> | pass@k |    1     |    2     |    4     |    8     |    16    |    32    |    64    |   128    |   Avg    |
> | :----: | :------: | :------: | :------: | :------: | :------: | :------: | :------: | :------: | :------: |
> | No reg |   44.7   |   56.2   | **65.5** | **72.2** | **76.9** | **79.6** |   81.0   |   82.2   |   69.8   |
> | IB reg | **46.1** | **56.5** |   65.4   |   71.7   |   76.2   |   79.4   | **81.6** | **82.8** | **70.0** |
>
> - Pass@k scores of Qwen3-14B-Base trained with PPO on AIME25:
> | pass@k |    1     |    2     |    4     |    8     |    16    |    32    |    64    |   128    |   Avg    |
> | :----: | :------: | :------: | :------: | :------: | :------: | :------: | :------: | :------: | :------: |
> | No reg | **34.8** | **41.6** |   47.9   |   53.2   |   57.6   |   61.5   |   64.8   |   67.4   |   53.6   |
> | IB reg |   33.2   |   41.5   | **49.9** | **57.8** | **64.9** | **71.3** | **77.5** | **81.6** | **59.7** |
>
> - Pass@k scores of Qwen3-14B-Base trained with DAPO on AIME25:
> | pass@k |    1     |    2     |    4     |    8     |    16    |    32    |    64    |   128    |   Avg    |
> | :----: | :------: | :------: | :------: | :------: | :------: | :------: | :------: | :------: | :------: |
> | No reg |   34.2   |   41.0   |   46.9   |   52.3   |   57.1   |   62.0   |   67.4   |   73.0   |   54.2   |
> | IB reg | **39.3** | **46.2** | **52.5** | **58.2** | **63.5** | **67.5** | **71.0** | **73.9** | **59.0** |
>
> Moreover, we further evaluate the pass@1 performance on four additional benchmark tasks: MATH500, Olympiad, Minerva, and GSM8K. The corresponding results are provided in the tables below. Our IB regularization again shows consistent improvements over the baseline, yielding an average gain of about two points for both PPO ($55.8 \rightarrow 58.6$) and DAPO ($58.4 \rightarrow 60.3$).
>
> - Pass@1 scores of Qwen3-14B-Base trained with PPO
> | pass@1 |  AMC23   |  AIME24  |  AIME25  | MATH500  | Olympiad | Minerva  |  GSM8K   |   Avg    |
> | :----: | :------: | :------: | :------: | :------: | :------: | :------: | :------: | :------: |
> | No reg |   81.5   | **45.4** | **34.8** |   86.0   |   55.9   |   32.4   |   54.9   |   55.8   |
> | IB reg | **82.8** |   44.7   |   33.2   | **88.0** | **58.9** | **34.6** | **67.8** | **58.6** |
>
> - Pass@1 scores of Qwen3-14B-Base trained with DAPO
> | pass@1 |  AMC23   |  AIME24  |  AIME25  | MATH500  | Olympiad | Minerva  |  GSM8K   |   Avg    |
> | :----: | :------: | :------: | :------: | :------: | :------: | :------: | :------: | :------: |
> | No reg |   83.7   |   44.7   |   34.2   |   86.4   |   58.0   | **37.5** |   64.5   |   58.4   |
> | IB reg | **86.9** | **46.1** | **39.3** | **88.2** | **58.5** |   33.8   | **69.3** | **60.3** |
>
> We have updated the paper accordingly, and the additional results can be found in Tables 2 and 3 of the revised version.

---

> ### Author Response · Authors · 2025-11-26
> **To Reviewer baqE (2/2)**
>
> **Q2:** *The improvements are relatively small (around +2 points) and may not be statistically significant. Figure 1 also suggests that the choice of training step or checkpoint may play a more substantial role in performance variation than the proposed regularization.*
>
> **A2:** Thanks. We respectfully argue that an average improvement of about two points on avg@32 (or pass@1) is substantial in the RLVR setting, where pass@1 directly reflects accuracy and gains are typically difficult to obtain. Although Figure 1 shows fluctuations during training, our evaluation uses both the top-1 and top-10 checkpoints to ensure a fair and stable comparison. Moreover, the performance curves of our IB regularization remain consistently higher than the baselines in most cases, indicating that the improvement is reliable rather than due to checkpoint variance.
>
> &nbsp;
>
> **Q3:** *How well does IB regularization generalize to non-math reasoning tasks?*
>
> **A3:** Thanks for your comments. We have further evaluated our method on the larger Qwen3-14B-Base model across additional benchmarks; please see **A1** for details. Our main experiments focus on math reasoning because it is the most widely adopted and reliable domain for assessing LLM reasoning capability in the RLVR setting.
>
> &nbsp;
>
>
> **Q4:** *How to empirically confirm that token advantage aligns with informational importance?*
>
> **A4:** Thanks. By definition in RL, token advantage captures the token-level credit assignment, thereby naturally reflecting token importance.
>
> &nbsp;
>
>
> **Q5:** *How robust is the method to different $\beta$ and $\alpha$ settings?*
>
> **A5:** Thanks. We first note that $\beta$ appears only in the theoretical derivation, while the practical IB regularizer uses a single coefficient $\alpha$. Since LLM RL experiments are highly computing-intensive, a full hyperparameter sensitivity study is challenging. Nonetheless, **all of our experiments, including the supplementary Qwen3-14B-Base results, use the same $\alpha=0.005$ without task specific tuning**. The consistent improvements across models and settings indicate the robustness of our approach.
>
> &nbsp;
>
>
> **Q6:** *What are the expected effects when scaling to larger LLMs? Scaling?*
>
> **A6:** Thanks for your comments. We have applied our method to the larger Qwen3-14B-Base model, and IB regularization continues to provide stable and consistent improvements over the No-reg baseline. Please refer to **A1** for the detailed results.

---

### Official Review · Reviewer_MYBT · 2025-11-01

**Soundness:** 3
**Presentation:** 3
**Contribution:** 2
**Rating:** 4
**Confidence:** 3

**Summary:**

The paper applies the Information Bottleneck principle to design a lightweight regularization for RL with verifiable rewards (RLVR). The method is computationally cheap and easy to integrate (one-line change in the loss function). Experiments indicate that the proposed regularization term improves LLM reasoning performance compared to training without regularization and to standard entropy regularization on mathematical benchmarks (AMC-23, AIME-24, AIME-25).

**Strengths:**

* The core idea is well motivated and intuitive.

* The modification adds no extra computation and is extremely simple to implement (one-line change), which is important for practical adoption since many new methods are hard to implement and deploy.

* The work provides a generalization bound with proofs for the proposed IBRO loss (Theorem 2).

**Weaknesses:**

* The experimental scope is quite limited. The regularization seems to be broadly applicable beyond mathematics, yet evaluation is restricted to three very similar math benchmarks. Broader testing on more diverse reasoning tasks (code, common-sense, logic, etc.) would better evaluate the improvement from the proposed regularization.

* The link between the IB-based theory and actual regularization term is unclear. It is not evident how the specific regularization term (lines 266–267, and Listing 1) follows from the surrogate IBRO objective (Lines 205-207). The rationale for replacing the modulation coefficient $\lambda_t$ with the advantage $A_t$ is hand-wavy, non-obvious (Lines 263-269). Additionaly, Assumption 1 (lines 197–199) is also questionable: it treats policy entropy as constant during RL training and is used in Theorem 1, yet Figure 2 shows substantial entropy changes under PPO and DAPO relative to initial values.

* Mathematical presentation and notation need refinement.  For example, on line 127 the PPO objective uses $r_t$ as the importance sampling ratio, while later $r$ denotes a random variable for the reasoning trajectory. Theorem 1 (lines 203–207) is stated without a proof. Although it seems to follow directly from the formula on line 200 and Assumption 1 (line 197), a written proof in appendix would be preferable. Equations in the main text are not numbered, which complicates referencing.

**Questions:**

4. **Questions**

* If Assumption 1 does not hold and the entropy of reasoning trajectories changes during training, as suggested by Figure 2, how can the success of the proposed regularization in Listing 1 be explained under the IB principle?
* The motivation on lines 268–269 that the term $A_t H_t$ encourages higher entropy for critical tokens and lower entropy for tokens with small or negative advantage seems counterintuitive. Shouldn’t critical reasoning tokens be generated with lower randomness, while stylistic tokens can vary more? Moreover, this rationale appears only loosely connected to the IBRO surrogate loss $\ell^t_{\text{IB}}$ on line 258. A low $\ell^t_{\text{IB}}$ for critical tokens does not by itself imply higher $H(o_t \mid q, o_{<t})$, since $\ell^t_{\text{IB}}$ also depends on $H(o_t \mid q, a, o_{<t})$, which could be the main driver of the low surrogate loss for such tokens.

---

> ### Author Response · Authors · 2025-11-26
> **To Reviewer MYBT (1/3)**
>
> **Q1:** *The experimental scope is quite limited. The regularization seems to be broadly applicable beyond mathematics, yet evaluation is restricted to three similar math benchmarks.*
>
> **A1:** We conducted additional experiments covering more models, evaluation tasks, and metrics. In particular, we performed RL post-training on the Qwen3-14B-Base model using both PPO and DAPO. The pass@k results on AMC23, AIME24, and AIME25, comparing our IB regularization with the No-reg baseline, are presented below. As shown, IB reg delivers stable and consistent improvements across all values of k.
>
> - Pass@k scores of Qwen3-14B-Base trained with PPO on AMC23:
> | pass@k | 1        | 2        | 4        | 8        | 16       | 32       | 64       | 128      | Avg      |
> | :----: | -------- | -------- | -------- | -------- | -------- | -------- | -------- | -------- | -------- |
> | No reg | 81.5     | 88.9     | 92.2     | 94.0     | 95.2     | 96.4     | 97.5     | 98.7     | 93.1     |
> | IB reg | **82.8** | **89.9** | **93.5** | **95.8** | **97.6** | **98.6** | **99.4** | **99.9** | **94.7** |
>
> - Pass@k scores of Qwen3-14B-Base trained with DAPO on AMC23:
> | pass@k |    1     |    2     |    4     |    8     |    16    |    32    |    64    |   128   |   Avg    |
> | :----: | :------: | :------: | :------: | :------: | :------: | :------: | :------: | :-----: | :------: |
> | No reg |   83.7   |   90.5   |   93.8   |   95.7   |   97.1   |   97.9   |   98.5   |  99.0   |   94.5   |
> | IB reg | **86.9** | **92.7** | **95.1** | **96.4** | **98.5** | **98.6** | **99.6** | **100** | **95.6** |
>
> - Pass@k scores of Qwen3-14B-Base trained with PPO on AIME24:
> | pass@k |    1     |    2     |    4     |    8     |    16    |    32    |    64    |   128    |   Avg    |
> | :----: | :------: | :------: | :------: | :------: | :------: | :------: | :------: | :------: | :------: |
> | No reg | **45.4** |   55.8   |   63.8   |   69.3   |   73.0   |   75.7   |   78.1   |   80.6   |   67.7   |
> | IB reg |   44.7   | **56.0** | **64.7** | **71.2** | **76.1** | **79.4** | **81.1** | **82.1** | **69.4** |
>
> - Pass@k scores of Qwen3-14B-Base trained with DAPO on AIME24:
> | pass@k |    1     |    2     |    4     |    8     |    16    |    32    |    64    |   128    |   Avg    |
> | :----: | :------: | :------: | :------: | :------: | :------: | :------: | :------: | :------: | :------: |
> | No reg |   44.7   |   56.2   | **65.5** | **72.2** | **76.9** | **79.6** |   81.0   |   82.2   |   69.8   |
> | IB reg | **46.1** | **56.5** |   65.4   |   71.7   |   76.2   |   79.4   | **81.6** | **82.8** | **70.0** |
>
> - Pass@k scores of Qwen3-14B-Base trained with PPO on AIME25:
> | pass@k |    1     |    2     |    4     |    8     |    16    |    32    |    64    |   128    |   Avg    |
> | :----: | :------: | :------: | :------: | :------: | :------: | :------: | :------: | :------: | :------: |
> | No reg | **34.8** | **41.6** |   47.9   |   53.2   |   57.6   |   61.5   |   64.8   |   67.4   |   53.6   |
> | IB reg |   33.2   |   41.5   | **49.9** | **57.8** | **64.9** | **71.3** | **77.5** | **81.6** | **59.7** |
>
> - Pass@k scores of Qwen3-14B-Base trained with DAPO on AIME25:
> | pass@k |    1     |    2     |    4     |    8     |    16    |    32    |    64    |   128    |   Avg    |
> | :----: | :------: | :------: | :------: | :------: | :------: | :------: | :------: | :------: | :------: |
> | No reg |   34.2   |   41.0   |   46.9   |   52.3   |   57.1   |   62.0   |   67.4   |   73.0   |   54.2   |
> | IB reg | **39.3** | **46.2** | **52.5** | **58.2** | **63.5** | **67.5** | **71.0** | **73.9** | **59.0** |
>
> Moreover, we further evaluate the pass@1 performance on four additional benchmark tasks: MATH500, Olympiad, Minerva, and GSM8K. The corresponding results are provided in the tables below. Our IB regularization again shows consistent improvements over the baseline, yielding an average gain of about two points for both PPO ($55.8 \rightarrow 58.6$) and DAPO ($58.4 \rightarrow 60.3$).
>
> - Pass@1 scores of Qwen3-14B-Base trained with PPO
> | pass@1 |  AMC23   |  AIME24  |  AIME25  | MATH500  | Olympiad | Minerva  |  GSM8K   | Avg  |
> | :----: | :------: | :------: | :------: | :------: | :------: | :------: | :------: | :--: |
> | No reg |   81.5   | **45.4** | **34.8** |   86.0   |   55.9   |   32.4   |   54.9   | 55.8 |
> | IB reg | **82.8** |   44.7   |   33.2   | **88.0** | **58.9** | **34.6** | **67.8** | **58.6** |
>
> - Pass@1 scores of Qwen3-14B-Base trained with DAPO
> | pass@1 |  AMC23   |  AIME24  |  AIME25  | MATH500  | Olympiad | Minerva  |  GSM8K   |   Avg    |
> | :----: | :------: | :------: | :------: | :------: | :------: | :------: | :------: | :------: |
> | No reg |   83.7   |   44.7   |   34.2   |   86.4   |   58.0   | **37.5** |   64.5   |   58.4   |
> | IB reg | **86.9** | **46.1** | **39.3** | **88.2** | **58.5** |   33.8   | **69.3** | **60.3** |
>
> We have updated the paper accordingly, and the additional results can be found in Tables 2 and 3 of the revised version.

---

> ### Author Response · Authors · 2025-11-26
> **To Reviewer MYBT (2/3)**
>
> **Q2.1:** *The link between the IB-based theory and actual regularization term is unclear. It is not evident how the specific regularization term (lines 266–267, and Listing 1) follows from the surrogate IBRO objective (Lines 205-207). The rationale for replacing the modulation coefficient with the advantage is hand-wavy, non-obvious (Lines 263-269).*
>
> **A2.1:** Thanks for your suggestion. We have revised the paper and added further explanation on using advantage as an intuitive proxy for the IBRO objective. Concretely, we reformulize the IBRO objective as
>
> $$
> \ell_{\text{IB}}^{t}
> = \beta H(o_t \mid o_{<t}, \boldsymbol{q}, \boldsymbol{a}) - H(o_t \mid o_{<t}, \boldsymbol{q})
> = \left(\beta - 1 - \beta  \frac{ I(\boldsymbol{a} ; o_t \mid o_{<t}, \boldsymbol{q}) }
>       { H(o_t \mid o_{<t}, \boldsymbol{q}) }\right) H_t
> $$
>
> The ratio $\frac{I\left(\boldsymbol{a} ; o_t \mid o_{<t}, \boldsymbol{q}\right)}{
>   H\left(o_t \mid o_{<t}, \boldsymbol{q}\right)
> } \in [0, 1]$ is a token-wise normalized conditional mutual information that measures how strongly \(o_t\) is related to the ground-truth answer \(\boldsymbol{a}\). On the other hand, the token advantage $A_t = A(o_t ; o_{<t}, \boldsymbol{q})$ captures the token-level credit assignment. Motivated by the intuition that more valuable tokens should be more strongly correlated with the ground truth, we model the normalized mutual information as a monotone function of the advantage
> $$
> \frac{I\left(\boldsymbol{a} ; o_t \mid o_{<t}, \boldsymbol{q}\right)}{H\left(o_t \mid o_{<t}, \boldsymbol{q}\right)}
> = \kappa(A_t),
> $$
> where $\kappa$ is monotone increasing in $A_t$. By reparameterizing with $\tilde{\kappa}(A_t) := \beta \kappa(A_t) - ({\beta - 1})$, we can rewrite $\ell_\texttt{IB}^t
> = -\tilde{\kappa}(A_t)\, H_t$. This leads to the following practical proxy for the IBRO objective:
> $$
> \min \ \mathcal{L}_{\texttt{IB}} = \sum_t^T -  \tilde{\kappa}(A_t) H_t \quad\Longleftrightarrow\quad \max \ \mathcal{J} = \sum_t^T \tilde{\kappa}(A_t)\ H_t
> $$
> In our implementation, we instantiate $\tilde{\kappa}$ with a simple affine increasing function $\tilde{\kappa}(A_t) =  \alpha A_t + \lambda $, which yields the desired proxy with advantages. More expressive forms of $\tilde{\kappa}$ can also be used, and exploring such variants is an interesting direction for future work.
>
> We have updated the paper accordingly, and the detail can be found in Lines 246 - 285 of the revised version.
>
> &nbsp;
>
> **Q2.2:** *Additionaly, Assumption 1 (lines 197–199) is also questionable: it treats policy entropy as constant during RL training and is used in Theorem 1, yet Figure 2 shows substantial entropy changes under PPO and DAPO relative to initial values*
>
> **A2.2:** Thanks for your comments. In the RLVR setting, the model optimizes the conditional distribution $\pi(r \mid q)$ given a training prompt $q$. Thus, **the entropy in RLVR literature is typically the conditional entropy $H(r \mid q)$, whereas Assumption 1 concerns the stability of the marginal (vanilla) entropy $H(r)$ throughout training**. To assess this assumption, we use the DAPO-Math-17k-Qwen3-235B-A22B-Thinking CoT dataset as $r$ and analyze both $H(r)$ and $H(r \mid q)$ during RL training on Qwen2.5-7B with DAPO. The trajectories of both quantities are shown in Figure 4 of the updated paper, and for clarity we also report their values at training Steps 0, 50, 100, 150, and 200 in the table below. As illustrated,**$H(r \mid q)$ decreases rapidly once training begins, while $H(r)$ remains stable, thereby supporting Assumption 1**. We have revised the paper accordingly, and additional details are provided in Appendix F.
>
> |              | Step 0 | Step 50 | Step 100 | Step 150 | Step 200 |
> | :----------: | :----: | :-----: | :------: | :------: | :------: |
> | $H(r\mid q)$ |  0.56  |  0.51   |   0.18   |   0.12   |   0.13   |
> |    $H(r)$    |  0.57  |  0.56   |   0.56   |   0.57   |   0.56   |
>
>
> &nbsp;
>
>
> **Q3:** *Mathematical presentation and notation need refinement. (1) The PPO objective uses $r_t$
>  as the importance sampling ratio, while later $r$ denotes a random variable for the reasoning trajectory. (2) Theorem 1 is stated without a proof.*
>
> **A3:** Thanks and addressed. (1) We have replaced $r_t$ with $s_t$ to denote the importance sampling ratio. (2) We have added a complete proof of Theorem 1 in Appendix A of the revised paper.
>
> &nbsp;
>
> **Q4:** *If Assumption 1 does not hold and the entropy of reasoning trajectories changes during training, as suggested by Figure 2, how can the success of the proposed regularization in Listing 1 be explained under the IB principle?*
>
> **A4:** Please refer to **A2.2**.

---

> ### Author Response · Authors · 2025-11-26
> **To Reviewer MYBT (3/3)**
>
> **Q5:** *(1) The motivation on lines 268–269 that the term $A_t H_t$ encourages higher entropy for critical tokens and lower entropy for tokens with small or negative advantage seems counterintuitive. Shouldn’t critical reasoning tokens be generated with lower randomness, while stylistic tokens can vary more? (2) Moreover, this rationale appears only loosely connected to the IBRO surrogate loss $\ell_{IB}^t$ on line 258. A low $\ell_{IB}^t$ for critical tokens does not by itself imply higher $H(o_t \mid q, o_{<t})$, since $\ell_{IB}^t$ also depends on $H(o_t \mid q, a, o_{<t})$, which could be the main driver of the low surrogate loss for such tokens.*
>
> **A5:** Thank you for the thoughtful comments.
>
> (1) We respectfully argue that **maintaining higher entropy for certain reasoning-related tokens (e.g., “wait,” “however,” “unless,” “thus”) can actually be beneficial for exploration in LLM RL post-training. Such tokens often signal branching points in the reasoning process, and allowing more variability at these steps increases the likelihood of sampling diverse solution paths, including correct ones**, which can then be reinforced. This intuition is consistent with empirical findings such as Figure 2(b) in [1]. In contrast, stylistic or low-information tokens (e.g., *is/am/are/a/the*) primarily serve coherence rather than reasoning. Reducing their entropy helps stabilize the trajectory and prevents unnecessary forking, thereby avoiding inefficient exploration.
>
> (2) We respectfully note that we do not claim that critical tokens should exhibit high $H(o_t \mid q, o_{<t})$ alone. **Under our IBRO framework, critical tokens are characterized by **both** *higher* $H(o_t \mid q, o_{<t})$ for better exploration and *lower* $H(o_t \mid q, a, o_{<t})$ for enhancing the succesful exploration or slution**. Therefore, our IBRO framework is quite reasonable and intuitive.
>
> &nbsp;
>
> [1] "Beyond the 80/20 rule: High-entropy minority tokens drive effective reinforcement learning for llm reasoning." *NeurIPS 2025*.

---

### Official Review · Reviewer_K4tL · 2025-11-01

**Soundness:** 2
**Presentation:** 3
**Contribution:** 2
**Rating:** 4
**Confidence:** 4

**Summary:**

This paper introduces information bottleneck (IB)-aware reasoning optimization (IBRO), aiming to maximize informativeness of reasoning trajectories with respect to the correct answer while minimizing reliance on prompt-specific details. The method derives a token-level surrogate for the IB objective and provide an efficient advantage-weighted entropy regularization method that can be incorporated into standard RL-based LLM post-training pipelines with minimal modification. Empirical evaluation across several mathematical reasoning benchmarks (AMC23, AIME24, AIME25) and RL algorithms (PPO, DAPO) demonstrates improvements over baseline and naive entropy regularization.

**Strengths:**

The manuscript is clearly written, explores both theoretical underpinnings and practical implications.
The method is highly efficient and practical: It requires negligible computational overhead and can be implemented with just one line of code modification.

**Weaknesses:**

1.The experiments in this paper are insufficiently comprehensive, having been conducted only on QWEN 2.5-7B. The universality of the method needs to be validated across different series of models.

2. Lack analysis of the hyperparameter α.

3.Assumption 1 (Page 4) that $\pi(\boldsymbol{r})$ remains invariant during post-training is asserted but not supported by empirical analysis or theoretical guarantee. In practice, LLM output distributions may shift as post-training progresses, potentially weakening the theoretical foundation. Can you give evidence (e.g., ablation or empirical distributional drift analysis) that this assumption holds?
I am happy to increase my score if these issues can be addressed.

**Questions:**

Generally, the response format is <think> </think> <answer> </answer>, with its length primarily determined by the reasoning process. If the length decreases due to an increase in the [EOS] probability, does this imply that the model did not output the final answer but instead directly output [EOS] during the reasoning process?

---

> ### Author Response · Authors · 2025-11-26
> **To Reviewer K4tL (1/2)**
>
> **Q1:** *The experiments are insufficiently comprehensive, having been conducted only on Qwen2.5-7B. The universality of the method needs to be validated across different series of models.*
>
> **A1:** We conducted additional experiments covering more models, evaluation tasks, and metrics. In particular, we performed RL post-training on the Qwen3-14B-Base model using both PPO and DAPO. The pass@k results on AMC23, AIME24, and AIME25, comparing our IB reg with the No-reg baseline, are presented below. As shown, IB reg delivers stable and consistent improvements across all values of k.
>
> - Pass@k scores of Qwen3-14B-Base trained with PPO on AMC23:
> | pass@k | 1        | 2        | 4        | 8        | 16       | 32       | 64       | 128      | Avg      |
> | :----: | -------- | -------- | -------- | -------- | -------- | -------- | -------- | -------- | -------- |
> | No reg | 81.5     | 88.9     | 92.2     | 94.0     | 95.2     | 96.4     | 97.5     | 98.7     | 93.1     |
> | IB reg | **82.8** | **89.9** | **93.5** | **95.8** | **97.6** | **98.6** | **99.4** | **99.9** | **94.7** |
>
> - Pass@k scores of Qwen3-14B-Base trained with DAPO on AMC23:
> | pass@k |    1     |    2     |    4     |    8     |    16    |    32    |    64    |   128   |   Avg    |
> | :----: | :------: | :------: | :------: | :------: | :------: | :------: | :------: | :-----: | :------: |
> | No reg |   83.7   |   90.5   |   93.8   |   95.7   |   97.1   |   97.9   |   98.5   |  99.0   |   94.5   |
> | IB reg | **86.9** | **92.7** | **95.1** | **96.4** | **98.5** | **98.6** | **99.6** | **100** | **95.6** |
>
> - Pass@k scores of Qwen3-14B-Base trained with PPO on AIME24:
> | pass@k |    1     |    2     |    4     |    8     |    16    |    32    |    64    |   128    |   Avg    |
> | :----: | :------: | :------: | :------: | :------: | :------: | :------: | :------: | :------: | :------: |
> | No reg | **45.4** |   55.8   |   63.8   |   69.3   |   73.0   |   75.7   |   78.1   |   80.6   |   67.7   |
> | IB reg |   44.7   | **56.0** | **64.7** | **71.2** | **76.1** | **79.4** | **81.1** | **82.1** | **69.4** |
>
> - Pass@k scores of Qwen3-14B-Base trained with DAPO on AIME24:
> | pass@k |    1     |    2     |    4     |    8     |    16    |    32    |    64    |   128    |   Avg    |
> | :----: | :------: | :------: | :------: | :------: | :------: | :------: | :------: | :------: | :------: |
> | No reg |   44.7   |   56.2   | **65.5** | **72.2** | **76.9** | **79.6** |   81.0   |   82.2   |   69.8   |
> | IB reg | **46.1** | **56.5** |   65.4   |   71.7   |   76.2   |   79.4   | **81.6** | **82.8** | **70.0** |
>
> - Pass@k scores of Qwen3-14B-Base trained with PPO on AIME25:
> | pass@k |    1     |    2     |    4     |    8     |    16    |    32    |    64    |   128    |   Avg    |
> | :----: | :------: | :------: | :------: | :------: | :------: | :------: | :------: | :------: | :------: |
> | No reg | **34.8** | **41.6** |   47.9   |   53.2   |   57.6   |   61.5   |   64.8   |   67.4   |   53.6   |
> | IB reg |   33.2   |   41.5   | **49.9** | **57.8** | **64.9** | **71.3** | **77.5** | **81.6** | **59.7** |
>
> - Pass@k scores of Qwen3-14B-Base trained with DAPO on AIME25:
> | pass@k |    1     |    2     |    4     |    8     |    16    |    32    |    64    |   128    |   Avg    |
> | :----: | :------: | :------: | :------: | :------: | :------: | :------: | :------: | :------: | :------: |
> | No reg |   34.2   |   41.0   |   46.9   |   52.3   |   57.1   |   62.0   |   67.4   |   73.0   |   54.2   |
> | IB reg | **39.3** | **46.2** | **52.5** | **58.2** | **63.5** | **67.5** | **71.0** | **73.9** | **59.0** |
>
> Moreover, we further evaluate the pass@1 performance on four additional benchmark tasks: MATH500, Olympiad, Minerva, and GSM8K. The results are provided in the tables below. Our IB regularization again shows consistent improvements over the baseline, yielding an average gain of two points for both PPO ($55.8 \rightarrow 58.6$) and DAPO ($58.4 \rightarrow 60.3$).
>
> - Pass@1 scores of Qwen3-14B-Base trained with PPO
> | pass@1 |  AMC23   |  AIME24  |  AIME25  | MATH500  | Olympiad | Minerva  |  GSM8K   |   Avg    |
> | :----: | :------: | :------: | :------: | :------: | :------: | :------: | :------: | :------: |
> | No reg |   81.5   | **45.4** | **34.8** |   86.0   |   55.9   |   32.4   |   54.9   |   55.8   |
> | IB reg | **82.8** |   44.7   |   33.2   | **88.0** | **58.9** | **34.6** | **67.8** | **58.6** |
>
> - Pass@1 scores of Qwen3-14B-Base trained with DAPO
> | pass@1 |  AMC23   |  AIME24  |  AIME25  | MATH500  | Olympiad | Minerva  |  GSM8K   |   Avg    |
> | :----: | :------: | :------: | :------: | :------: | :------: | :------: | :------: | :------: |
> | No reg |   83.7   |   44.7   |   34.2   |   86.4   |   58.0   | **37.5** |   64.5   |   58.4   |
> | IB reg | **86.9** | **46.1** | **39.3** | **88.2** | **58.5** |   33.8   | **69.3** | **60.3** |
>
> We have updated the paper accordingly, and the additional results can be found in Tables 2 and 3 of the revised version.

---

> > ### Comment · Reviewer_K4tL · 2025-11-28
> >
> > Thanks for the updated results on the Qwen3-14B model. They are pretty good augmentation to the original findings.

---

> ### Author Response · Authors · 2025-11-26
> **To Reviewer K4tL (2/2)**
>
> **Q2:** *Lack analysis of the hyperparameter $\alpha$.*
>
> **Q2:** Thanks. Since LLM RL experiments are highly computing-intensive, a full hyperparameter sensitivity study is challenging. Nonetheless, **all of our experiments, including the supplementary Qwen3-14B-Base results, use the same $\alpha=0.005$ without task specific tuning**. The consistent improvements across models and settings indicate the robustness of our approach.
>
> &nbsp;
>
> **Q3:** *Assumption 1 (Page 4) that remains invariant during post-training is asserted but not supported by empirical analysis or theoretical guarantee. In practice, LLM output distributions may shift as post-training progresses, potentially weakening the theoretical foundation. Can you give evidence (e.g., ablation or empirical distributional drift analysis) that this assumption holds? I am happy to increase my score if these issues can be addressed.*
>
> **A3:** Thanks for your comments. In the RLVR setting, the model optimizes the conditional distribution $\pi(r \mid q)$ given a training prompt $q$. Thus, **the entropy in RLVR literature is typically the conditional entropy $H(r \mid q)$, whereas Assumption 1 concerns the stability of the marginal (vanilla) entropy $H(r)$ throughout training**. To assess this assumption, we use the DAPO-Math-17k-Qwen3-235B-A22B-Thinking CoT dataset as $r$ and analyze both $H(r)$ and $H(r \mid q)$ during RL training on Qwen2.5-7B with DAPO. The trajectories of both quantities are shown in Figure 4 of the updated paper, and for clarity we also report their values at training Steps 0, 50, 100, 150, and 200 in the table below. As illustrated, **$H(r \mid q)$ decreases rapidly once training begins, while $H(r)$ remains stable, thereby supporting Assumption 1**. We have revised the paper accordingly, and additional details are provided in Appendix F.
>
> |              | Step 0 | Step 50 | Step 100 | Step 150 | Step 200 |
> | :----------: | :----: | :-----: | :------: | :------: | :------: |
> | $H(r\mid q)$ |  0.56  |  0.51   |   0.18   |   0.12   |   0.13   |
> |    $H(r)$    |  0.57  |  0.56   |   0.56   |   0.57   |   0.56   |
>
> &nbsp;
>
> **Q4:** *Generally, the response format is [object Object] [object Object] [object Object] [object Object], with its length primarily determined by the reasoning process. If the length decreases due to an increase in the [EOS] probability, does this imply that the model did not output the final answer but instead directly output [EOS] during the reasoning process?*
>
> **A4:** Thank you for the comment. We agree that “an increased [EOS] probability truncating the final answer” is an interesting angle for interpreting the correlation between response length and accuracy. While this analysis is beyond the scope of our paper, we hypothesize that the relationship between response length and model performance is more nuanced. Rather than directly terminating the response by emitting [EOS] prematurely, a gradual increase in [EOS] probability may encourage the model to produce answers more hastily, without sufficient reasoning or verification, which in turn leads to degraded performance.

---

> > ### Comment · Reviewer_K4tL · 2025-11-28
> >
> > Thanks for the thorough response and the additional experiment on verifying the constancy of H(r). I think the theoretical basis is now more solid. I will raise my score.

---

### Official Review · Reviewer_WcRr · 2025-11-03

**Soundness:** 2
**Presentation:** 4
**Contribution:** 2
**Rating:** 4
**Confidence:** 3

**Summary:**

The paper proposes IB-aware Reasoning Optimization (IBRO), grounding LLM reasoning in the information bottleneck (IB) principle, and derives a practical IB regularization term that multiplies token entropy by token advantages and adds it to standard RL objectives. The method is claimed to be a one-line change, with pseudocode shown, and is evaluated on AMC23/AIME24/AIME25 under PPO and DAPO, yielding small but consistent avg@32 gains and more stable entropy dynamics/length profiles.

**Strengths:**

[Originality]
A crisp, unifying viewpoint: formalizing “good CoT” as minimizing dependence on prompt-specific details while maximizing informativeness about the correct answer (IBRO) and turning this into a token-level surrogate and advantage-weighted entropy regularizer.

[Quality]
Clear derivation from IBRO to a token-level objective and a practical approximation that requires only quantities already computed by PPO/GRPO-style RL (token entropy and advantages).
The “one-line change” is explicit (Listing 1). Results show consistent though modest improvements in avg@32 across three math benchmarks and two RL settings, plus interpretable plots for entropy and response length.

[Clarity]
Writing and structure are strong; key assumptions (e.g., treating H(r)H(r) as constant) and the IB surrogate objective (Theorem 1) are presented cleanly.

[Significance]
 Low-overhead regularization that slots into existing RLVR pipelines is practically useful for groups already running PPO/GRPO-style training.

**Weaknesses:**

1. Practical novelty is limited (IB-on-tokens).
While the IB framing is elegant, the implementable contribution (advantage-weighted entropy) is close to known entropy/density-shaping ideas; the main novelty is the IB interpretation and the advantage weighting rather than a fundamentally new mechanism. Please position against recent entropy-minimization/entropy-shaping/diversity-aware policy works with a small comparison table (objective, where entropy acts, when it increases/decreases, and reported effects).

2. IB feasibility/faithfulness.
The mutual information terms among $r$, $q$, $a$ are not measured; they’re upper-bounded/approximated via entropy surrogates plus assumptions (e.g., treating $𝐻(𝑟)$ constant; bounding $𝐻(𝑜_𝑡\mid 𝑞, 𝑎)$ before mapping to advantage-weighted entropy. Please (i) quantify approximation error or (ii) at least correlate the proposed $𝐿_{𝐼𝐵}$ proxy with held-out accuracy during training to empirically support the IB interpretation.

3. Theory is largely incremental.
Appendix A reuses a known generalization framework (their Theorem 3 is Theorem 2 of Kawaguchi et al., 2023) and brings it into the present notation; the new step is substituting the IBRO-style loss $𝐿_{𝐼𝐵}$​ and relating it to $\|\delta\theta\|$. This is a solid contextualization but not a new bound. Please state this clearly and emphasize the paper’s value as a principled application of IB theory to RLVR rather than a theoretical breakthrough.

4. Baselines are narrow.
Only PPO and DAPO are considered; all are RLVR. To support broader claims about “reasoning,” please add at least one non-RL baseline (e.g., SFT/CoT-SFT, DPO), or show that adding IB-reg to a non-RL finetuning objective also helps. If compute is tight, a smaller-scale ablation on GSM8K with LoRA would still be informative.

5. Benchmarks/metrics are narrow.
All three datasets are math-only (AMC23/AIME24/AIME25), and the primary metric is avg@32. Add 1–2 broader reasoning sets (e.g., GSM8K, BBH, MATH-500) and include at least pass@k/top-1 and a length-controlled metric to rule out length confounds.

6. Ablations are thin.
Current ablations only show entropy dynamics and mean lengths. Please include:
i) Hyperparameter sensitivity for \alpha and \beta;
ii) Remove advantage weighting (plain entropy reg) to isolate its contribution;
iii) Try sign-only advantages (positive vs negative) to test the mechanism posited for DAPO/GRPO;
iv) A short compute overhead table (wall-clock and FLOPs) to substantiate “negligible cost.”

7. Missing qualitative case studies.
Include side-by-side CoTs (success and failure) showing how IB-reg changes intermediate steps (e.g., earlier pruning of spurious paths or delayed [EOS] as hypothesized). This would ground the mechanism beyond aggregate metrics.

8. Reproducibility/finetuning details are underspecified.
It’s unclear whether training is full-model or LoRA/PEFT, and the exact compute profile is missing. Please clarify parameter-efficiency, optimizer state sharding, gradient checkpointing, reference policy handling (they say no KL to a reference), and batch/throughput numbers; this is critical for reproducing RLVR at 7B. The appendices list many hparams, but the finetuning mode (full vs PEFT) and hardware/runtime are not explicit.

**Questions:**

1. Faithfulness of IB surrogate:
Could you provide empirical evidence (e.g., correlation analysis) that the proposed surrogate
$𝐿_{𝐼𝐵}$ meaningfully reflects mutual information among $𝑟$, $𝑞$, $𝑎$ or tracks accuracy improvements during training?

2. Novelty of the theoretical result:
Theorem 3 appears to restate an existing IB generalization bound (Kawaguchi et al., 2023) with minimal change. Could you clarify what new theoretical insight or justification your adaptation provides?

3. Baselines and generalization:
The experiments only include PPO and DAPO. Do you expect IB regularization to benefit non-RL fine-tuning methods (e.g., SFT, DPO, process supervision)?

4. Evaluation scope:
The benchmarks are all math reasoning datasets with avg@32 as the main metric. Could you test on an additional reasoning domain (e.g., GSM8K, BBH) or include other metrics such as pass@k?

5. Qualitative evidence:
Would you consider adding example reasoning traces to show whether IB regularization changes CoT structure, e.g., fewer hallucinations or better intermediate logic?

6. Training setup clarification:
Please clarify whether fine-tuning was full-model or parameter-efficient (e.g., LoRA). This detail is essential for reproducibility and understanding computational cost.

---

> ### Author Response · Authors · 2025-11-26
> **To Reviewer WcRr (1/4)**
>
> **Q1:** *Practical novelty is limited (IB-on-tokens). The implementable contribution (advantage-weighted entropy) is close to known entropy/density-shaping ideas.*
>
> **A1:** Thanks. Our IB regularization is derived from an Information Bottleneck perspective. Although the final expression reduces to a neat advantage-weighted entropy term, we respectfully note that this formulation and its IB-driven motivation have not been explored in either conventional RL or current LLM post-training work.
>
> &nbsp;
>
> **Q2:** *The mutual information terms among $q$, $r$, $a$ are not measured; they’re upper-bounded/approximated via entropy surrogates plus assumptions.*
>
> **A2:** Thanks for your comments. Because the mutual information term is typically intractable, we upper-bound it with an entropy-based surrogate under Assumption 1, which concerns the stability of the marginal (vanilla) entropy $H(r)$ during training. In contrast, the entropy typically discussed in the RLVR literature is the conditional entropy $H(r \mid q)$. To assess this assumption, we use the DAPO-Math-17k-Qwen3-235B-A22B-Thinking CoT dataset as $r$ and analyze both $H(r)$ and $H(r \mid q)$ during RL training on Qwen2.5-7B with DAPO. The trajectories of both quantities are shown in Figure 4 of the updated paper, and for clarity we also report their values at training Steps 0, 50, 100, 150, and 200 in the table below. As illustrated, $H(r \mid q)$ decreases rapidly once training begins, while $H(r)$ remains stable, thereby supporting Assumption 1. We have revised the paper accordingly, and additional details are provided in Appendix F.
>
> |              | Step 0 | Step 50 | Step 100 | Step 150 | Step 200 |
> | :----------: | :----: | :-----: | :------: | :------: | :------: |
> | $H(r\mid q)$ |  0.56  |  0.51   |   0.18   |   0.12   |   0.13   |
> |    $H(r)$    |  0.57  |  0.56   |   0.56   |   0.57   |   0.56   |
>
>
> &nbsp;
>
>
> **Q3:** *Theorem 1 of generalization bound is incremental. Please state this clearly and emphasize the paper’s value as a principled application of IB theory to RLVR rather than a theoretical breakthrough.*
>
> **A3:** Thanks for your suggestion. Theorem 1 applies Kawaguchi’s result within our IBRO framework to provide better theoretical insight, and it does not affect the subsequent method. We have moved it to the Appendix in the revised version to keep the main presentation focused on our approach.
>
> &nbsp;
>
> **Q4:** *Baselines are narrow. Only PPO and DAPO are considered; all are RLVR. To support broader claims about “reasoning,” please add at least one non-RL baseline.*
>
> **A4:** Thanks. RL is the most popular and widely used approach for enhancing LLM reasoning without relying on costly CoT data. In particular, RLVR enables effective improvement using only rule-based verifiable answers, making it an ideal testbed for evaluating LLM RL algorithms. Moreover, PPO and DAPO are representative methods spanning both critic-based and critic-free paradigms. Our IB regularization consistently improves reasoning performance under both, demonstrating the effectiveness and generality of our approach.

---

> ### Author Response · Authors · 2025-11-26
> **To Reviewer WcRr (2/4)**
>
> **Q5:** *Benchmarks/metrics are narrow. All three datasets are math-only (AMC23/AIME24/AIME25), and the primary metric is avg@32.*
>
> **A5:** Thank you for the suggestion. During the rebuttal period, we conducted additional experiments covering more models, evaluation tasks, and metrics. In particular, we performed RL post-training on the Qwen3-14B-Base model using both PPO and DAPO. The pass@k results on AMC23, AIME24, and AIME25, comparing our IB reg with the No-reg baseline, are presented below. As shown, IB reg delivers stable and consistent improvements across all values of k.
>
> - Pass@k scores of Qwen3-14B-Base trained with PPO on AMC23:
> | pass@k | 1        | 2        | 4        | 8        | 16       | 32       | 64       | 128      | Avg      |
> | :----: | -------- | -------- | -------- | -------- | -------- | -------- | -------- | -------- | -------- |
> | No reg | 81.5     | 88.9     | 92.2     | 94.0     | 95.2     | 96.4     | 97.5     | 98.7     | 93.1     |
> | IB reg | **82.8** | **89.9** | **93.5** | **95.8** | **97.6** | **98.6** | **99.4** | **99.9** | **94.7** |
>
> - Pass@k scores of Qwen3-14B-Base trained with DAPO on AMC23:
> | pass@k |    1     |    2     |    4     |    8     |    16    |    32    |    64    |   128   |   Avg    |
> | :----: | :------: | :------: | :------: | :------: | :------: | :------: | :------: | :-----: | :------: |
> | No reg |   83.7   |   90.5   |   93.8   |   95.7   |   97.1   |   97.9   |   98.5   |  99.0   |   94.5   |
> | IB reg | **86.9** | **92.7** | **95.1** | **96.4** | **98.5** | **98.6** | **99.6** | **100** | **95.6** |
>
> - Pass@k scores of Qwen3-14B-Base trained with PPO on AIME24:
> | pass@k |    1     |    2     |    4     |    8     |    16    |    32    |    64    |   128    |   Avg    |
> | :----: | :------: | :------: | :------: | :------: | :------: | :------: | :------: | :------: | :------: |
> | No reg | **45.4** |   55.8   |   63.8   |   69.3   |   73.0   |   75.7   |   78.1   |   80.6   |   67.7   |
> | IB reg |   44.7   | **56.0** | **64.7** | **71.2** | **76.1** | **79.4** | **81.1** | **82.1** | **69.4** |
>
> - Pass@k scores of Qwen3-14B-Base trained with DAPO on AIME24:
> | pass@k |    1     |    2     |    4     |    8     |    16    |    32    |    64    |   128    |   Avg    |
> | :----: | :------: | :------: | :------: | :------: | :------: | :------: | :------: | :------: | :------: |
> | No reg |   44.7   |   56.2   | **65.5** | **72.2** | **76.9** | **79.6** |   81.0   |   82.2   |   69.8   |
> | IB reg | **46.1** | **56.5** |   65.4   |   71.7   |   76.2   |   79.4   | **81.6** | **82.8** | **70.0** |
>
> - Pass@k scores of Qwen3-14B-Base trained with PPO on AIME25:
> | pass@k |    1     |    2     |    4     |    8     |    16    |    32    |    64    |   128    |   Avg    |
> | :----: | :------: | :------: | :------: | :------: | :------: | :------: | :------: | :------: | :------: |
> | No reg | **34.8** | **41.6** |   47.9   |   53.2   |   57.6   |   61.5   |   64.8   |   67.4   |   53.6   |
> | IB reg |   33.2   |   41.5   | **49.9** | **57.8** | **64.9** | **71.3** | **77.5** | **81.6** | **59.7** |
>
> - Pass@k scores of Qwen3-14B-Base trained with DAPO on AIME25:
> | pass@k |    1     |    2     |    4     |    8     |    16    |    32    |    64    |   128    |   Avg    |
> | :----: | :------: | :------: | :------: | :------: | :------: | :------: | :------: | :------: | :------: |
> | No reg |   34.2   |   41.0   |   46.9   |   52.3   |   57.1   |   62.0   |   67.4   |   73.0   |   54.2   |
> | IB reg | **39.3** | **46.2** | **52.5** | **58.2** | **63.5** | **67.5** | **71.0** | **73.9** | **59.0** |
>
> Moreover, we further evaluate the pass@1 performance on four additional benchmark tasks: MATH500, Olympiad, Minerva, and GSM8K. The results are provided in the tables below. Our IB regularization again shows consistent improvements over the baseline, yielding an average gain of two points for both PPO ($55.8 \rightarrow 58.6$) and DAPO ($58.4 \rightarrow 60.3$).
>
> - Pass@1 scores of Qwen3-14B-Base trained with PPO
> | pass@1 |  AMC23   |  AIME24  |  AIME25  | MATH500  | Olympiad | Minerva  |  GSM8K   |   Avg    |
> | :----: | :------: | :------: | :------: | :------: | :------: | :------: | :------: | :------: |
> | No reg |   81.5   | **45.4** | **34.8** |   86.0   |   55.9   |   32.4   |   54.9   |   55.8   |
> | IB reg | **82.8** |   44.7   |   33.2   | **88.0** | **58.9** | **34.6** | **67.8** | **58.6** |
>
> - Pass@1 scores of Qwen3-14B-Base trained with DAPO
> | pass@1 |  AMC23   |  AIME24  |  AIME25  | MATH500  | Olympiad | Minerva  |  GSM8K   |   Avg    |
> | :----: | :------: | :------: | :------: | :------: | :------: | :------: | :------: | :------: |
> | No reg |   83.7   |   44.7   |   34.2   |   86.4   |   58.0   | **37.5** |   64.5   |   58.4   |
> | IB reg | **86.9** | **46.1** | **39.3** | **88.2** | **58.5** |   33.8   | **69.3** | **60.3** |
>
> We have updated the paper accordingly, and the additional results can be found in Tables 2 and 3 of the revised version.

---

> ### Author Response · Authors · 2025-11-26
> **To Reviewer WcRr (3/4)**
>
> **Q6.1**: *Ablations are thin. Current ablations only show entropy dynamics and mean lengths. Please include: i) Hyperparameter sensitivity for $\alpha$ and $\beta$.*
>
> **A6.1:** Thanks. We first note that $\beta$ appears only in the theoretical derivation, while the practical IB regularizer uses a single coefficient $\alpha$. Since LLM RL experiments are highly computing-intensive, a full hyperparameter sensitivity study is challenging. Nonetheless, **all of our experiments, including the supplementary Qwen3-14B-Base results, use the same $\alpha=0.005$ without task specific tuning**. The consistent improvements across models and settings indicate the robustness of our approach.
>
> &nbsp;
>
> **Q6.2:** *ii) Remove advantage weighting (plain entropy reg) to isolate its contribution;*
>
> **A6.2:** Thanks. We have conducted the "plain entropy reg" experiments in our paper, which we call "Naive reg", Please refer to Table 1 and Figure 1. As shown in our experiments, "Naive reg" has lower performance than "No reg".
>
> &nbsp;
>
>
> **Q6.3:** *iii) Try sign-only advantages (positive vs negative) to test the mechanism posited for DAPO/GRPO;*
>
> **A6.3:** Thanks. Because all token advantages are same among the sequence for GRPO-like algorithms, our IB reg is very similar to use the sign of advantages as regularization coefficient under GRPO/DAPO, while different sequences/responses possess different positive or negative value advantages. Due to the limited computing resources, we will continue conduct this experiments after evaluating our apporach on more models and tasks.
>
> &nbsp;
>
>
> **Q6.4:** *iv) A short compute overhead table (wall-clock and FLOPs) to substantiate “negligible cost.”*
>
> **A6.4:** Thank you for the suggestion. Since both the advantages and token probabilities required by IB regularization are already available in standard LLM RL pipelines, our method introduces negligible additional computational cost compared with the No-reg baseline. To empirically confirm this, we report the FLOPs and training wall-clock time in the table below. TFLOPs is measured as the average per training step over 800 steps, and the wall-clock time reflects the total end-to-end training duration. All experiments are conducted on 32 NVIDIA H20 GPUs. As shown, IB reg and the No-reg baseline exhibit nearly identical computational overhead, supporting our claim that the additional cost is negligible. We have updated the paper accordingly, and further details can be found in Table 5 of the revised version.
>
> - Computational overhead of Qwen3-14B-Base on PPO
> | Method | TFLOPs | Wall-Clock Time (hours) |
> | :----: | :----: | :---------------------: |
> | No reg |  61.3  |          75.1           |
> | IB reg |  62.3  |          71.6           |
>
> - Computational overhead of Qwen3-14B-Base on DAPO
> | Method | TFLOPs | Wall-Clock Time (hours) |
> | :----: | :----: | :---------------------: |
> | No reg |  99.4  |          31.3           |
> | IB reg | 100.7  |          32.5           |
>
> &nbsp;
>
>
> **Q7:** *Missing qualitative case studies. Include side-by-side CoTs (success and failure) showing how IB-reg changes intermediate steps (e.g., earlier pruning of spurious paths or delayed [EOS] as hypothesized). This would ground the mechanism beyond aggregate metrics.*
>
> **A7:** Thanks for your comments. Our apprach is measured over thousands of questions, which we believe provide a more reliable assessment than a small number of hand-picked qualitative cases. Although qualitative examples can be illustrative, they tend to be intuitive and subjective. The consistent gains achieved by our IB regularization across multiple models and tasks offer strong evidence of its effectiveness.
>
> &nbsp;
>
>
> **Q8:** *Reproducibility/finetuning details are underspecified. It’s unclear whether training is full-model or LoRA/PEFT, and the exact compute profile is missing.*
>
> **A8:** Thank you for the question. All RLVR experiments in our paper use full-model tuning, which allows us to fully leverage the capacity of the pretrained model and push its performance limits. At the same time, our IB regularization is fully compatible with LoRA and other PEFT methods. We provide all hyperparameter settings in Appendix C.

---

> ### Author Response · Authors · 2025-11-26
> **To Reviewer WcRr (4/4)**
>
> **Q9:** *Faithfulness of IB surrogate: Could you provide empirical evidence (e.g., correlation analysis) that the proposed surrogate $L_{IB}$ meaningfully reflects mutual information among $r,q,a$, or tracks accuracy improvements during training?*
>
> **A9:** Please refer to **A2**.
>
> &nbsp;
>
> **Q10:** *Novelty of the theoretical result: Theorem 3 appears to restate an existing IB generalization bound (Kawaguchi et al., 2023) with minimal change. Could you clarify what new theoretical insight or justification your adaptation provides?*
>
> **A10:** Please refer to **A3**.
>
> &nbsp;
>
> **Q11:** *Baselines and generalization: The experiments only include PPO and DAPO. Do you expect IB regularization to benefit non-RL fine-tuning methods (e.g., SFT, DPO, process supervision)?*
>
> **A11:** Please refer to **A4**.
>
> &nbsp;
>
> **Q12:** *Evaluation scope: The benchmarks are all math reasoning datasets with avg@32 as the main metric. Could you test on an additional reasoning domain (e.g., GSM8K, BBH) or include other metrics such as pass@k?*
>
> **A12:** Please refer to **A5**.
>
> &nbsp;
>
> **Q13:** *Qualitative evidence: Would you consider adding example reasoning traces to show whether IB regularization changes CoT structure, e.g., fewer hallucinations or better intermediate logic?*
>
> **A13:** Please refer to **A7**.
>
> &nbsp;
>
> **Q14:** *Training setup clarification: Please clarify whether fine-tuning was full-model or parameter-efficient (e.g., LoRA). This detail is essential for reproducibility and understanding computational cost.*
>
> **A14:** Please refer to **A8**.

---

### Author Response · Authors · 2025-12-01
**Summary of Revisions**

The authors sincerely thank all reviewers for their insightful feedback and suggestions. We have extensively revised the paper to strengthen the empirical validation, theoretical grounding, and clarity. Below we summarize the main updates made in direct response to the reviews:

1. **Expanded empirical validation (Tables 2 & 3)**

   We added new experiments on Qwen3-14B-Base (under both PPO and DAPO) and evaluated performance across multiple metrics (pass@k) and seven benchmarks (AMC23, AIME24, AIME25, MATH500, Olympiad, Minerva, GSM8K). Our IB regularization consistently improves performance while maintaining negligible compute overhead.

&nbsp;

2. **Clarification and justification of Assumption 1 (Figure 4)**

   We conducted additional experiments showing that the marginal entropy $H(r)$, rather than the commonly used conditional entropy $H(r|q)$ in RLVR, remains stable throughout training. This supports Assumption 1.

&nbsp;

3. **Improved explanation of using token advantage as a surrogate for the IBRO objective (Lines 246-285)**

   We reformulated the IBRO objetive using the normalized mutual information term $\frac{I \left(\boldsymbol{a} ; o_t \mid o_{<t}, \boldsymbol{q}\right)}{H \left(o_t \mid o_{<t}, \boldsymbol{q}\right)}$ (Line 250), which quantifies how strongly each token relates to the ground truth answer. This quantity is reasonably approximated by an advantage-based function $\kappa(A_t)$. Consequently, using token advantages as a surrogate for the IBRO objective is both intuitive and well justified.

&nbsp;

4. **Compute overhead analysis (Table 5)**

   We added experiments confirming that IB regularization introduces negligible additional cost compared with the baseline in terms of FLOPs per step and end-to-end training time.

---

### Meta-Review · Area_Chair_Tm3P · 2026-01-08

**Summary:**

This paper proposes IB-aware Reasoning Optimization, applying the Information Bottleneck principle to LLM reasoning during RL post-training. The main contribution of this paper is to add a lightweight IB regularization term that multiplies token entropy by token advantages and adds it to standard RL objectives and only requires one-line code modification to existing RL pipelines.

The initial review of this paper is 4,4,4,4,6. The primary concerns centered on: (1) limited practical novelty beyond the IB interpretation, (2) insufficient empirical validation of theoretical assumptions, (3) narrow experimental scope, (4) modest improvement margins, (5) incremental theoretical contribution.

**Reviewer Concerns:**

Addressed concerns: narrow experimental scope is mostly addressed by one more model and some new benchmarks. Empirical validation of theoretical assumptions is also addressed in the new Figure 4. However, the limited novelty and incremental theoretical contribution still likely outstanding.

**Reviewer Scores:**

Reviewer K4tL 4 -> 6. Reviewer MYBT likely 4 -> 6. Reviewer baqE 50% = 4, 50% -> 6. Other 2 reviewers (score 4 and 6) likely unchanged.

---

### Decision · Program_Chairs · 2026-01-26

Reject